# Heterologous mogrosides biosynthesis in cucumber and tomato by genetic manipulation

Jingjing Liao[1,5], Tingyao Liu[2], Lei Xie[1], Changming Mo[3], Jing Qiao[1], Xiyang Huang[4], Shengrong Cui[1], Xunli Jia[1], Zuliang Luo [1✉] & Xiaojun Ma [1✉]

Mogrosides are widely used as high-value natural zero-calorie sweeteners that exhibit an array of biological activities and allow for vegetable flavour breeding by modern molecular biotechnology. In this study, we developed an In-fusion based gene stacking strategy for transgene stacking and a multi-gene vector harbouring 6 mogrosides biosynthesis genes and transformed it into *Cucumis sativus* and *Lycopersicon esculentum*. Here we show that transgenic cucumber can produce mogroside V and siamenoside I at 587 ng/g FW and 113 ng/g FW, respectively, and cultivated transgenic tomato with mogroside III. This study provides a strategy for vegetable flavour improvement, paving the way for heterologous biosynthesis of mogrosides.

[1] Institute of Medicinal Plant Development, Chinese Academy of Medical Sciences & Peking Union Medical College, Beijing 100193, China. [2] College of Horticulture, Shenyang Agricultural University, Shenyang 110866, China. [3] Guangxi Crop Genetic Improvement and Biotechnology Lab, Guangxi Academy of Agricultural Sciences, Nanning 530007, China. [4] Guangxi Key Laboratory of Plant Functional Phytochemicals and Sustainable Utilization, Guangxi Institute of Botany, Guangxi Zhuang Autonomous Region and Chinese Academy of Sciences, Guilin 541006, China. [5] Present address: The Artemisinin Research Center, Institute of Chinese Materia Medica, China Academy of Chinese Medical Sciences, Beijing 100700, China. ✉email: zuliangluo@163.com; xjma@implad.ac.cn

The preference for food flavour pervades the entire evolutionary history of humans, and this preference has occasionally been higher than human nutritional requirements have been. Flavours have played a pivotal role in food preference, and tastiness afforded a sense of pleasure, which has important implications for improving appetite, digestion and increasing the use rate of nutrients[1]. Moreover, humans have already begun improving vegetable and fruit flavours, which has ultimately led to the guidance of diet expenditure and improvement to human health[2]. Over the past half century, the breeding goal of high crop productivity has indirectly caused a reduction in flavour and nutrients[3]. Owing to genetic linkage, quality traits appear to be negatively related to yield, and desirable flavour, high yield and high quality are complex traits controlled by multiple genes[4]. In general, therefore, plant breeding for flavour enhancement remains a major challenge on the basis of the high yield and good quality. Since 1983, major advances in transgenic biotechnology have allowed breeding for enhanced flavour and nutritional quality to be more simplified and feasible[5]. As has been shown previously, there is an obvious absence of flavour-associated volatiles in modern commercial tomato cultivars. Consequently, scientists have modified tomato plants such that the fruits present a lemon flavour and a rose-like aroma via the expression of *Ocimum basilicum* geraniol synthase[6]. Furthermore, *TomLoxC* has been widely acknowledged to affect tomato fruit flavour by catalysing lipid-derived C5 and C6 volatile biosynthesis[7]. Moreover, previous research has indicated the transformation of *VpVAN* gene regulating the vanillin biosynthesis has changed the unique flavour of pepper[8]. Obviously, plant breeding involving homologous or heterologous transgenes associated with flavour has become a popular research topic likely to continue in the future.

Sweetness preference is a kind of human instinct, and sweetness has been described as a fundamental hedonic pleasure[9]. Sweet foods with sugar as the essential ingredient have markedly increased obesity, diabetes and cardiovascular diseases in all age groups worldwide[10]; thus, it is necessary to explore sugar-free sweeteners for use in daily diets. Equally, scientists transformed sweet-tasting proteins into cucumber[11], tomato[12,13], strawberry[14], pear[15], and lettuce[13,16] for sweet-taste crops breeding. However, sweet proteins were commonly limited by high price, insufficient supply, poor taste and short shelf life and stability[17]. In 1983, mogroside V, a non-sugar sweetener isolated from the unique *Siraitia grosvenorii* fruit (Cucurbitaceae; Luo-han-guo or monk fruit), was discovered in China[18–20], and were approved by FDA in 2010. Typically, mogrosides are divided into several sweet components, such as mogroside III, siamenoside I and mogroside V. The sweetness of mogrosides differ depending on the number of glycosylation and glycosylation sites[21]. The great advantage of mogroside V is its superb flavour compared with that of steviosides, rubusoside and glycyrrhizin, which normally are slightly bitter tasting sweet[22,23]. Interestingly, this natural sweetener with antiglycation effects[24] has a high sweetness, has a low-calorie content and is nontoxic; its sweetness is approximately 300 times higher than that of sucrose[25,26]; and its history of medicinal use is >300 years[27]. Mogrosides have been approved and used by all kinds of recognised brands at home and abroad, and it has been applied in the >4, 300 products all over the world by the end of September 2019. In China, mogrosides is thought as a kind of safe, non-toxic and natural sweetener, which is widely used in the food, beverage and pharmaceutical industries and has great commercial potential worldwide[27]. In 2011, our group investigated the mogroside V biosynthesis pathway, and 40 key enzyme-encoding genes were cloned for the first time[28], which provided donor genes for crop breeding for sweet flavour improvement. According to the literature, the precursor of mogroside V biosynthesis, 2,3-oxidosqualene, is extensively found in plants[29] (Supplementary Fig. S1). Therefore, it is vitally important that various mogroside V synthase genes be transformed into candidate plants to develop sweet plants, and these transgenic plants can also be used as promising materials for mogroside V production.

During the past 30 years, many transgenic plants for which a single trait has been genetically improved have been applied to promote commercial cultivation in some countries and areas[30]. With the development of metabolic engineering and synthetic biology, multigene transformation has been applied for regulating biosynthesis and improving multiple biological properties instead of single-gene transformation, which is an emerging trend in genetic breeding[31]. To date, there have been numerous breakthroughs in plants genetically engineered for the biofortification of micronutrients, phytonutrients and bioactive components, such as β-carotene-enriched Golden rice[32], potato[33], banana[34] and canola[35], anthocyanin-[36], L-DOPA-[37], folate[38] and flavonol-[39], betalain-biofortified[40] tomato fruits, which were developed via multigene transformation involving targeted metabolite biosynthesis pathways. To achieve the de novo synthesis of mogroside V, the key problem is the integration of 6 mogrosides biosynthesis genes into candidate plants. We therefore developed a simple and efficient multigene expression system based on In-fusion technology and self-cleaving 2A peptides. Moreover, 6 mogroside V synthase genes have been successfully introduced into cucumber and tomato, and all genes had high transcript levels in the transgenic plants. Accordingly, we developed sweet cucumber transgenic plant with mogroside V and slightly sweet tomato with mogroside III (MIII) firstly. This study describes extensive prospective application in the field of vegetable and fruit flavour breeding and provides a valuable and captivating blueprint for elaborately developing exceptional plant germplasms with certain characteristic and multiple flavours.

## Results

**Promoter activity assays**. The AtUBQ10 and AtPD7 promoters were isolated from *Arabidopsis thaliana* and ligated into a pBI121 vector together with the GUS reporter gene (Fig. 1a). To assess the suitability of the promoters, transient expression was performed in the cotyledons of *Cucumis sativus* as described previously[41]. Leaves of *Nicotiana benthamiana* and cotyledons of *Cucumis sativus* were infected with *Agrobacterium* harbouring pBI121 in which the GUS gene was driven by AtUBQ10, AtPD7 and CaMV 35S promoters under the same conditions. To characterize the function of these promoters, histochemical staining was performed as described previously[42]. As expected, there was a high level of GUS activity under the control of the AtUBQ10 and AtPD7 promoters in *Cucumis sativus* and *Nicotiana benthamiana* (Fig. 1b, c), and the highest expression level was detected at 5 d after infiltration of *Cucumis sativus*. Therefore, the ability of the AtUBQ10 and AtPD7 promoters to drive gene transcription in *Cucumis sativus* and *Nicotiana benthamiana* was comparable to that of the CaMV 35S promoter, and there was no significant difference between these promoters (Fig. 1b and Fig. 1c). These promoters could therefore be used as strong constitutive promoters to drive gene expression in the multigene vector.

**Design of multigene expression vector for mogrosides biosynthesis**. It has been widely demonstrated that the precursor of mogrosides biosynthesis is 2,3-oxidosqualene, which is synthesized through the mevalonate pathway and catalysed by a series of enzymes to synthesize mogroside V in *Siraitia grosvenorii* (Supplementary Fig. S1). Therefore, the binary plasmid pCAMBIA1300 harbouring the mogrosides synthesis-related

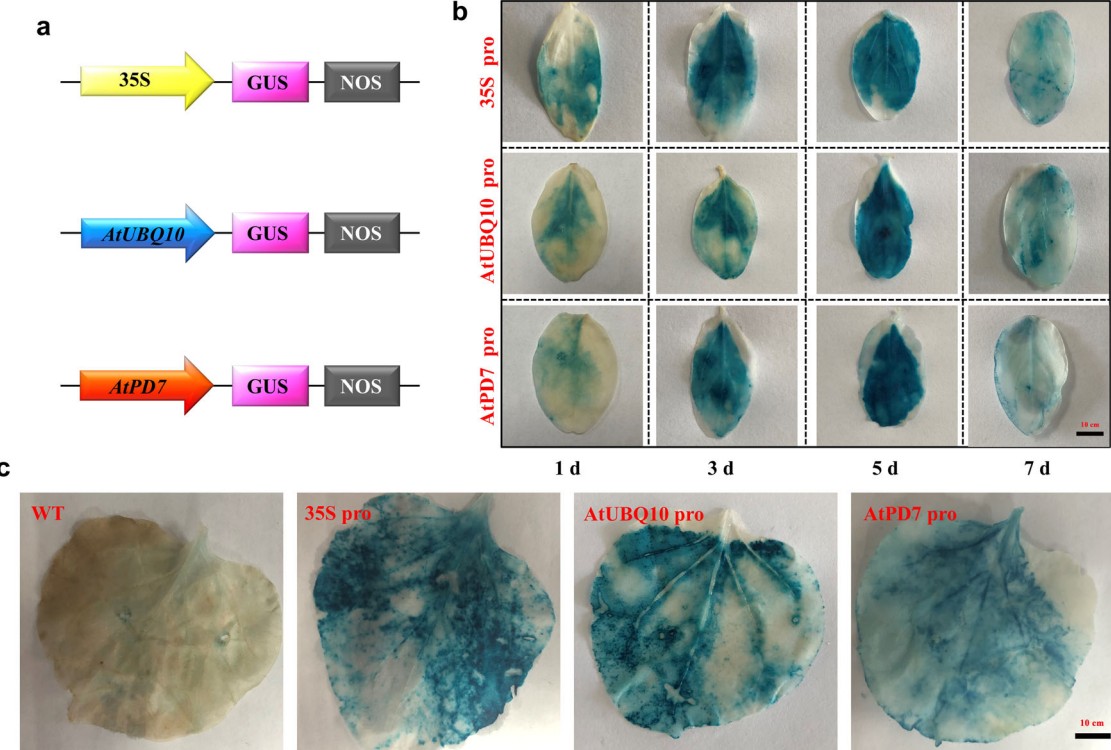

**Fig. 1 Cloning and analysis of candidate promoters in cucumber and tobacco. a** Recombinant plasmid with different promoters. **b** Histochemical GUS assay of transient expression in cucumber. **c** Histochemical GUS assay of transient expression was performed after 48 h of infiltration in tobacco. WT plants were used as negative controls. 35S pro: CaMV 35S promoter; AtUBQ10 pro: AtUBQ10 promoter; AtPD7 pro: AtPD7 promoter.

enzyme-encoding genes *SgSQE1, SgCS, SgEPH2, SgP450, SgUGT269-1* and *SgUGT289-3* together with Hyg resistance gene (Hyg, selection marker) driven by AtPD7, AtUBQ10 and CaMV 35S promoters was constructed via In-fusion technology and self-cleaving 2A peptides (Supplementary Fig. S2a). First, all the target genes were ligated into pBI121 or pCAMBIA1300 to produce the first gene expression cassette. Second, the region harbouring the promoter, target gene and terminator was cloned and ligated into pCAMBIA1300 to produce a double-gene expression cassette. Then, combining the first gene expression cassette and the double-gene expression cassette, we constructed a triple-gene expression cassette. Finally, the triple-gene expression cassettes were ligated into the final vector by P2A peptides. (Supplementary Fig. S2a). The corresponding U22p-SCE plasmid was identified via PCR (Supplementary Fig. S2b). This multigene expression vector was large (the length from the left border to the right border was ~21.5 kb) and was used to synthesize mogrosides. The U22p-SCE plasmid was introduced into *Agrobacterium tumefaciens* GV3101, and the transformants were further used for genetic transformation.

**Transient expression assays**. To further confirm the availability of the multigene vectors, a transient expression assay was performed in *Cucumis sativus*. All mogrosides biosynthesis-related genes were expressed in the cotyledons of *Cucumis sativus* via agroinfiltration. In accordance with the methods of our preliminary study, we sampled the cotyledons of *Cucumis sativus* at 5 d and the cotyledons of tobacco leaves at 48 h after infiltration for HPLC-ESI-MS/MS analysis. As indicated in the Supplementary Fig. S3a and S3b, MII-E and MIII accumulated slightly in the cotyledons of *Cucumis sativus* transformed with the U22p-SCE multigene expression vector, but MI-A1 was not detected. Unfortunately, no SI or MV was observed in the multiple

transient expression assay. The lack of accumulation of SI and MV suggested at least two possibilities: (i) the transient expression assay generally lasted for a short time, in this case, the accumulation of MIII substrate was inadequate for the production of SI and MV in the cotyledons of (*Cucumis sativus*), and (ii) the multigene expression vector was too large to suppress the gene expression, which resulted in the reduction of MII-E and MIII. Nevertheless, the transient expression assay confirmed that multigene expression vector was available. Thus, the stable transformation of multigene expression vectors is required for their genetic transformation into *Cucumis sativus* and *Lycopersicon esculentum*, which provides a new idea about mogrosides-accumulating vegetable breeding.

**Generation of mogrosides-accumulating cucumber**. To enhance the nutritional characteristics of cucumber, the multigene expression vector U22p-SCE was introduced into *Agrobacterium tumefaciens* strain GV3101 and the mogrosides biosynthetic pathway was genetically engineered to produce mogrosides in cucumber plants. About 500 cucumber leaf explants were subjected to *Agrobacterium*-mediated transformation with the U22p-SCE vector. Among these, the total number of Hyg-resistant lines was about 15, but many Hyg-resistant lines were not able to root, survive and grow on seedling. Finally, only 5 transgenic plants were grown in the green house after domestication and transplantation. All transgenic plants and non-transformed plants were cultivated in the greenhouse with the consistent growth environment (Fig. 2a). To detect the integration of the transgenes into the genome of the cucumber plants (Fig. 2b), genomic DNA was extracted to determine the presence of target genes and Hyg genes using gene-specific primers by PCR (Supplementary Table S4). Fragments of the expected size were detected in all Hyg-resistant transgenic lines (Fig. 2c). A 1587 bp fragment of

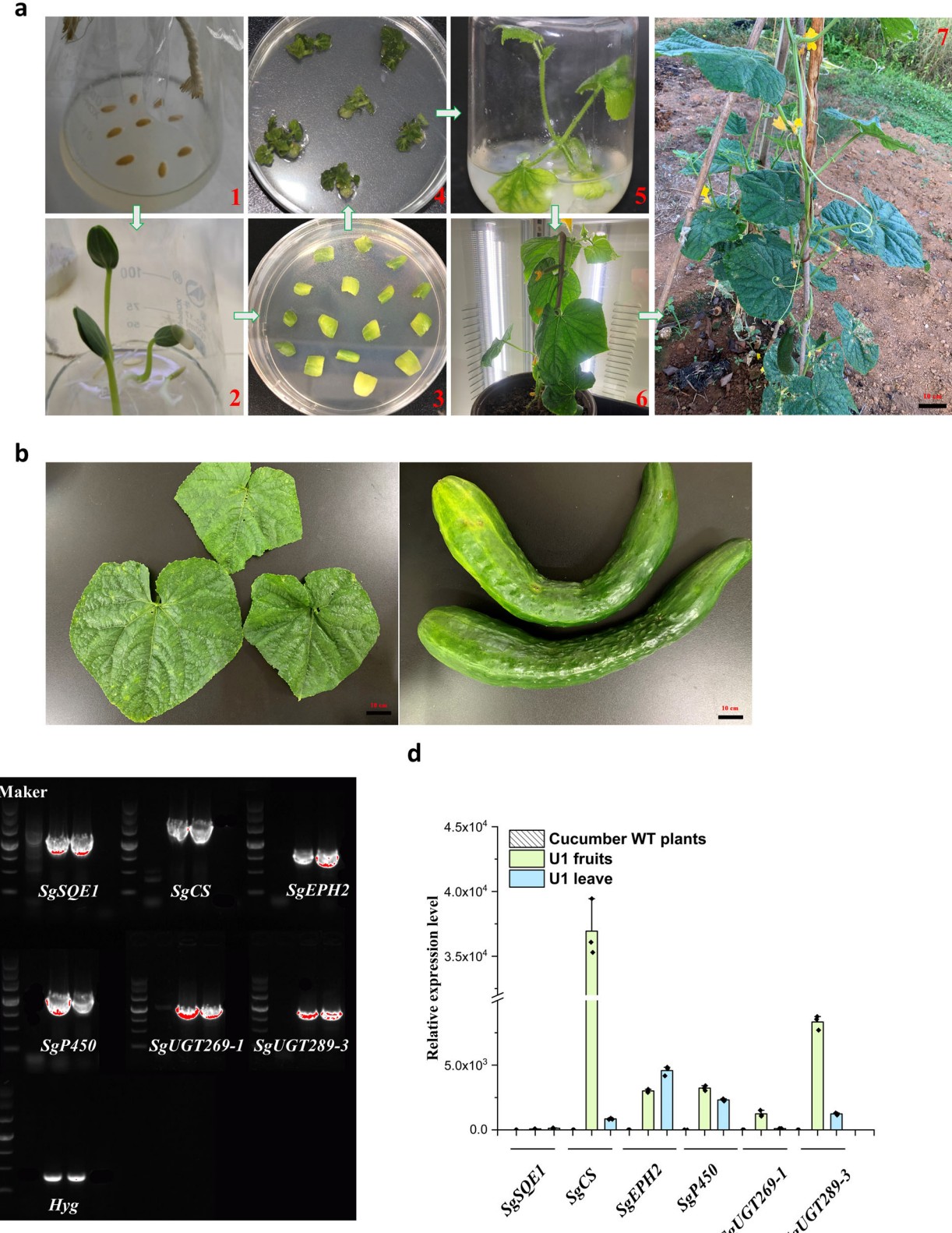

**Fig. 2 Genetic transformation of cucumber with *Agrobacterium* harbouring the U22p-SCE vector. a** Genetic transformation of cucumber with *Agrobacterium* harbouring the U22p-SCE vector. (1) Sterilized seeds. (2) 4-days-old seedlings. (3) The cotyledons were cut and used as explants. (4) The explants in the selected medium. (5) The regenerated plants in the rooting media. (6, 7) Regenerated plants. **b** Leaves and fruits of transgenic cucumber line U1. **c** PCR-based detection of U1. The lanes from left to right represent the Maker, WT, U1 fruits and U1 leaves. An image of the DNA marker (4.5 kb) is in the bottom right-hand corner of the figure. **d** Transcript level analysis of transgenic cucumber line U1 according to qRT-PCR. The *Csactin* is used as an internal control. Expression of cucumber WT plants was set to 1. The data are presented as the mean values ± SDs, $n = 3$ biologically independent samples.

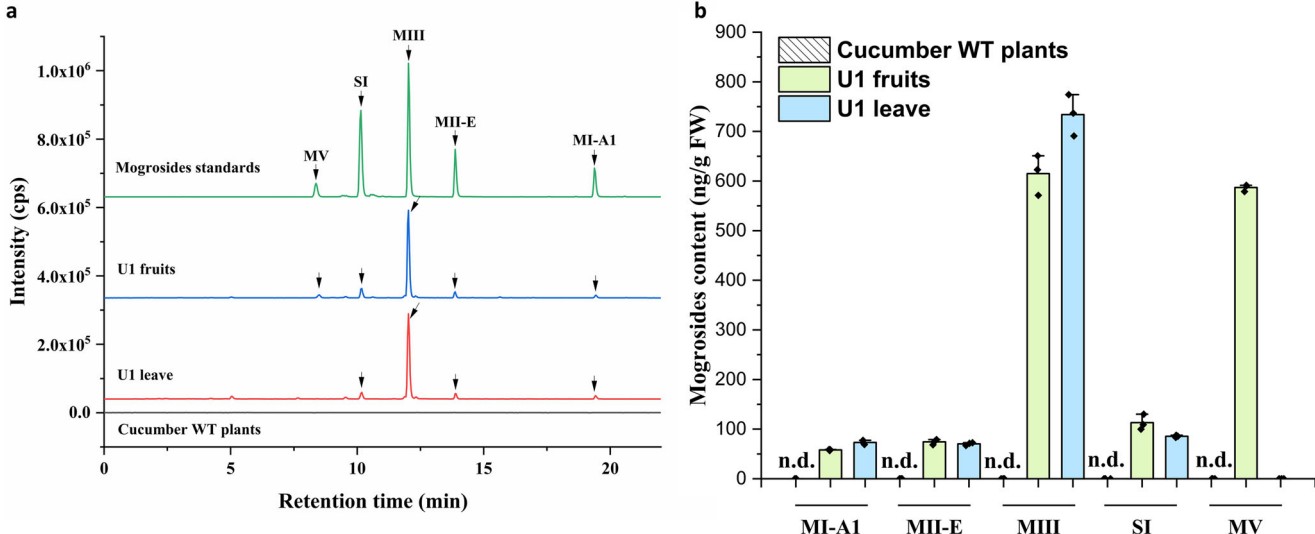

**Fig. 3 Production of mogrosides in transgenic cucumber line. a** HPLC-ESI-MS/MS analysis of mogrosides in transgenic cucumber line U1. The black arrows indicate the peak of mogrosides. **b** Accumulation of mogrosides in transgenic cucumber line U1. n.d., not detected. The data are presented as the mean values ± SDs, $n = 3$ biologically independent samples.

*SgSQE1*, 2280 bp fragment of *SgCS*, 951 bp fragment of *SgEPH2*, 1421 bp fragment of *SgP450*, 1253 bp fragment of *SgUGT269-1*, 1026 bp fragment of *SgUGT289-3* and 392 bp fragment of the Hyg resistance gene were simultaneously detected in the one transgenic plant (U1) but were not amplified from WT cucumber plants (Fig. 2c). Other plants almost never existed all six genes using PCR amplication. The data of the cucumber transformation experiment were listed in the Supplementary Table S1. In total, we obtained only one independent transgenic cucumber line after transformation with the U22p-SCE vector. Although the transformation frequency was relatively low in this study, this is the first study in which 6 genes were transformed simultaneously into the cucumber genome, and a large-vector transformation is unpredictable. Previous studies have suggested that the genetic transformation efficiency of cucumber is still highly variable and that the genotypes of explants that can be used are limited. To improve the transformation efficiency with multigene vector in an *Agrobacterium*-mediated system, the essential step in genetic transformation still need to optimized, which the key tasks and the targets for the next stage. We further measured the expression level of transgenes via qRT-PCR (Fig. 2d). Quantitative real-time PCR (qRT-PCR) analysis revealed that the *SgSQE1, SgCS, SgEPH2, SgP450, SgUGT269-1* and *SgUGT289-3* transcript levels were markedly higher in the WT plants. In this assay, the expression of 6 transgenes varied in the leaves and fruits. For example, in all six transgenes, the relatively lower *SgSQE* expression was detected in the both of leaves and fruits (Fig. 2d), which was located in the second gene position for 2A peptides construct driven by CaMV 35S promoter (Supplementary Fig. S2a). The gene expression level between the first and the second gene positions was commonly affected by the utilization of different 2A peptides[43]. All in all, PCR and qRT-PCR detection indicated that the structural genes involved in mogrosides biosynthesis were expressed in the transgenic lines of cucumber. Finally, only the U1 transgenic plant was acquired using *Agrobacterium*-mediated transformation.

On the basis of a molecular analysis, the composition and content of mogrosides in the transgenic cucumber line were determined using HPLC-ESI-MS/MS. Mogrol, MI-A1, MII-E, MIII, SI and MV standards were detected at retention times of 4.01, 19.38, 13.88, 12.03, 10.14 and .8.37 min, respectively (Fig. 3a and Supplementary Fig. S4a). As shown in Fig. 3b and

Supplementary Fig. S4b, mogrol, MIA-1, MII-E, MIII, and SI were detected in transgenic cucumber fruits and leaves, except for MV, which was found only in the fruits. MV accumulated to ~587 ng/g fresh weight (FW) in the fruits, and the contents of the other mogrosides is listed in Fig. 3b and Supplementary Fig. S4b. Mogrosides were not detected in the WT cucumber plants. Furthermore, the metabolites were further identified by UPLC-ESI-QTOF-MS/MS, and the total ion chromatographs in negative mode are shown in Fig. 4. According to the retention time, true molecular weight and mass spectrometry of standards, five mogrosides were observed in transgenic cucumber U1 fruits, including MI-A1, MII-E, MIII, SI and MV. And [M + HCOO-H]⁻ and [M-H]⁻ were commonly observed in the MS/MS spectrum at negative ion mode. The fracture mode of MV was as follows: the characteristic fragments easily obtained one molecule of formic acid, which resulted in product ions at $m/z$ 1331.5443 in MS spectrum. The deprotonated ions of [M-H]⁻ ($m/z$ 1285.5507) was clearly observed in the MS/MS spectrum (Supplementary Fig. S5a). Moreover, SI obtained one molecule of formic acid to show [M + HCOO-H]⁻ ($m/z$ 1169.5096) and generated a deprotonated ion [M-H]⁻ at $m/z$ 1123.5330 (Supplementary Fig. S5b). Similarly, based on MS/MS spectra, compounds 3, 4 and 5 were identified as MIII, MII-E and MI-A1 (Supplementary Fig. S5c–e). Notably, the presence of mogrosides confirmed that the simultaneous expression of the 6 genes was successful in the transgenic cucumber line. Sweet cucumber was generated successfully by reconstructing mogrosides biosynthesis in transgenic cucumber. Interesting, in the U1 leave, the presence of mogrosides was also proved to be a great potential for SI production using agricultural wastes. In our study, we developed a convenient and highly efficient multigene transformation strategy via In-fusion technology and 2A peptide linkers. These results indicate that this strategy could be applied in cucumber and have important implications for developing mogrosides-accumulating vegetables firstly.

**Generation of MIII-accumulating tomato.** Similarly, the multigene vector was transformed into Micro-Tom tomato. About 500 explants were transformed in the assay, and twenty Hyg-resistant tomato transgenic lines were generated by multiple transformation methods. Among them, only four lines had

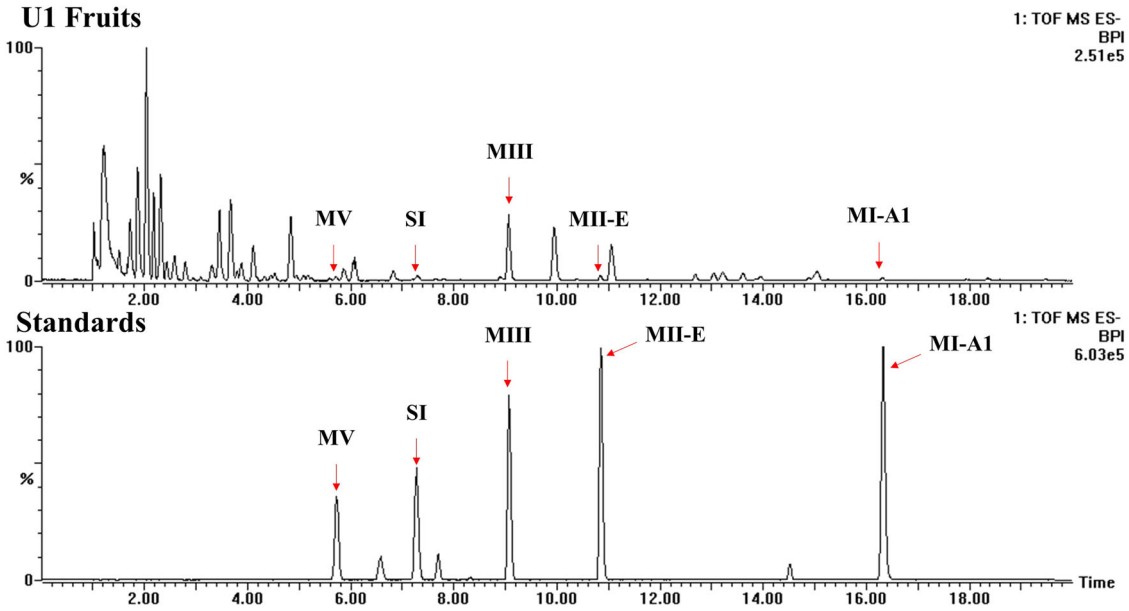

**Fig. 4 The total ion chromatogram of mogrosides in transgenic cucumber line U1 and standards.** MIA-1, MII-E, MIII, SI, and MV represented the mogroside I-A1, mogroside II-E, mogroside III, siamenoside I, and mogroside V, respectively.

mogrosides biosynthesis-related genes detected via PCR with specific primers for the *SgSQE1*, *SgCS*, *SgEPH2*, *SgP450*, *SgUGT269-1*, *SgUGT289-3* and Hyg genes. Fragments of the expected size were obtained from the genomic DNA of four candidate transgenic tomato lines. All the mogrosides biosynthesis-related genes were amplified from these four (S8, S10, S14 and S17) lines out of the twenty lines transformed with the U22p-SCE vector (Fig. 5a); none of the target genes were observed in the WT plants (Fig. 5b). That is, this multigene vector was successfully introduced in the tomato genome. To further confirm the transgenic lines, the expression level of the target genes was examined in the S8, S10, S14 and S17 lines via qPCR (Fig. 5c). All mogrosides biosynthesis-related genes were over-expressed, although there were relatively higher expression levels of all genes in the S10 line (Fig. 5c). No transcripts of the target genes were detected in the WT tomato plants, which indicated that 6 structural genes involved in mogrosides biosynthesis were expressed in transgenic tomato fruits. That is, the mogrosides pathway was successfully introduced into the transgenic tomato plants firstly.

The production of mogrosides in the four transgenic tomato fruits was measured by HPLC-ESI-MS/MS. The extracted ion chromatogram (EIC) of the HPLC suggested that a small amount of MIII accumulated in transgenic tomato line S10 (Fig. 6a), and the retention times were consistent with those of the mogrosides standards. No MIII was detected in the WT plants (Fig. 6a). The content of MIII was 25.92 ng/g FW (Fig. 6b), and we also found a small amount of MI-A1 (5.65 ng/g), MII-E (2.33 ng/g), SI and MV. However, the content of SI and MV is still below the limit of quantification (<LOQ) (Fig. 6a). The target compounds in tomato transgenic plants were further confirmed by UPLC-ESI-QTOF-MS/MS. The result has been shown in Supplementary Fig. S6a–f. In general, tomato transgenic plants also produced mogrosides, but the level was lower compared to mogrosides content in cucumber transgenic plants. As shown in Supplementary Fig. S4c and S4d, there were only minute amount of mogrol accumulation in the transgenic tomato lines S8 and S17. And the production of mogrol was absent in the Micro-Tom WT plants.

## Discussion

Cucumber is a commonly nutritious vegetable, yet it has a bland taste. Polyphenolic compounds and salivary proteins provide cucumbers with a subtle astringent taste, which causes some people to avoid eating cucumbers[44,45]. To improve the flavour of cucumber, *Siraitia grosvenorii* mogrosides biosynthesis genes were transformed into cucumber plants, which produces a germplasm with mogroside V. And in the fruits of transgenic cucumber line U1, mogrol, MI-A, MII-E, MIII, and SI were found and the content were 36.88, 58, 74.3, 615, and 113 ng/g FW, respectively. Although it is well known that mogroside IA-1, II-E are bitter-tasting glycosides, SI and MV are extremely sweet, MIII is tasteless or slightly sweet, which cultivate cucumbers to taste sweet, blend flavours, or totally new tastes to commonly known vegetables. And the content of MI-A and MII-E with bitter flavour were present at too low a level in the fruits of transgenic cucumber, and therefore the consequence for flavour of transgenic cucumber is relatively small. Similarly, the transformation of tomato produced a small amount of mogroside III in tomato. The content of SI and MV is still below the limit of quantification (<LOQ) in the transgenic tomato lines. The taste description of MIII is weak or tasteless compared with mogroside V[46], and it is reported in literature that mogroside III is 195 times sweeter than sucrose[47]. Previously its maltase inhibitory effect has been investigated previously[48], and MIII reduced pulmonary fibrosis and may have therapeutic potential for treating fibrosis[49]. Combining with the natural flavour of tomato fruits, it is presumed that the taste of transgenic tomato fruits is slightly sweet, which allow vegetables to taste better. In general, tomato transgenic plants also produced mogrosides, but the level was lower compared to mogrosides content in cucumber transgenic plants. The reason for this phenomenon might be a lack of precursor accumulation, stepwise glycosylation strategy or T-DNA insertion position. To reduce the consumption of substrates, gene-editing technology such as clustered, regularly interspaced, short palindromic repeats (CRISPR)/CRISPR-associated 9 (Cas9), which have widely used for the functional verification of key enzymes and regulatory elements[50], can be applied to knock out genes involved in branch pathways. Overexpression of upstream genes

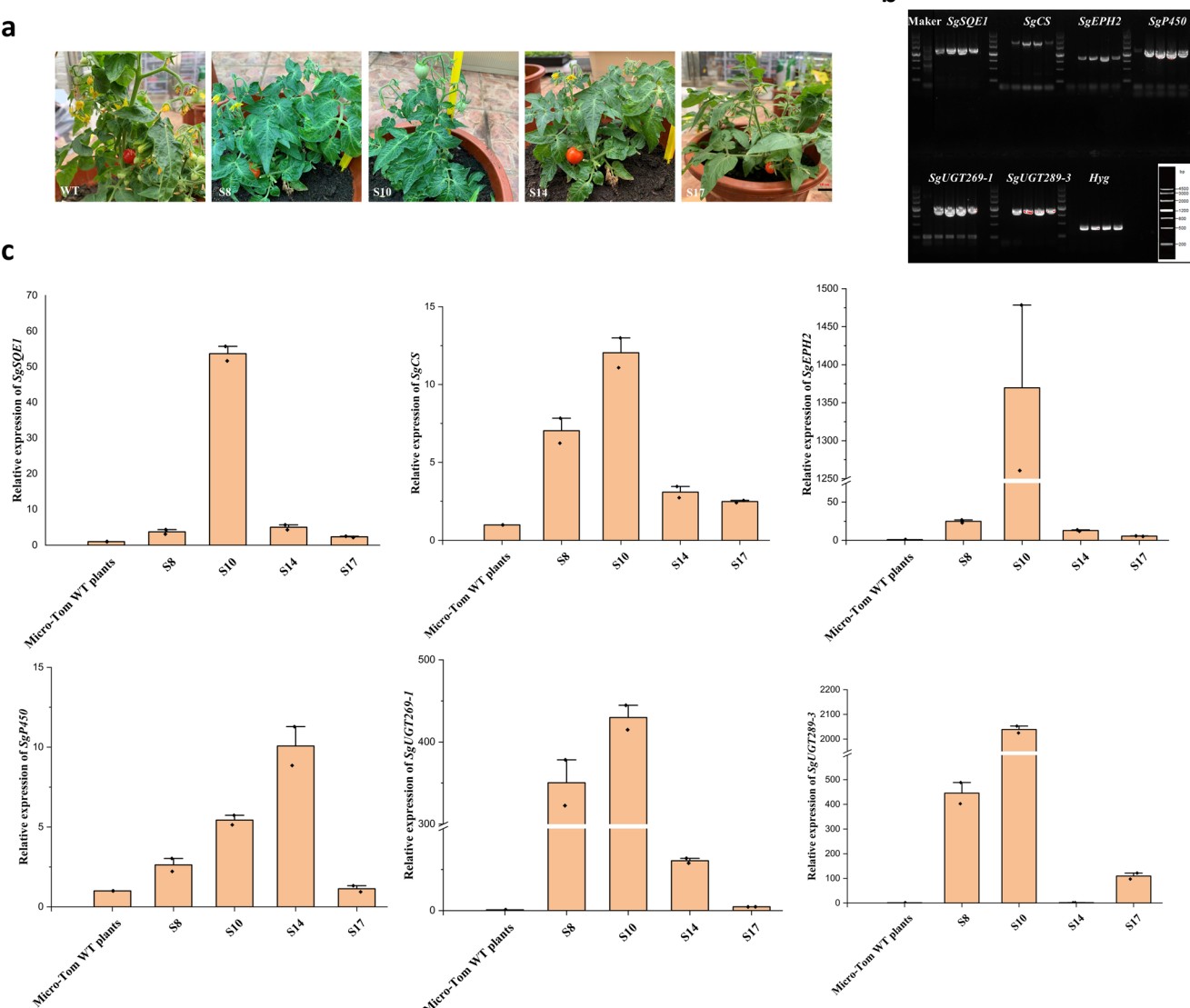

**Fig. 5 Molecular analysis and detection of mogrosides in transgenic tomato lines. a** Micro-Tom tomato wild-type plants (WT) and transgenic tomato plants. **b** PCR-based analysis of the transgenic tomato fruits. The lanes from left to right represent the Maker, WT, S8, S10, S14 and S17. An image of the DNA marker (4.5 kb) is in the bottom right-hand corner of the figure. **c** Relative expression level analysis of 6 mogrosides biosynthesis genes in transgenic tomato fruits. The *Leactin* is used as an internal control. Expression of tomato WT plants was set to 1. The data are presented as the mean values ± SDs, $n = 3$ biologically independent samples.

encoding various rate-limiting enzymes, such as HMGR (encoding 3-hydroxy-3-methyl glutaryl coenzyme A reductase) and SQS (encoding squalene synthase), is beneficial to the accumulation of substrates[51]. Despite this, 2, 3-Oxidosqualene is widely found in all kinds of plants, theoretically, it could be exploited to transfer mogrosides biosynthesis genes. Our study marks the success of the transformation of mogrosides biosynthesis genes into heterologous plants, which offers a fresh perspective on breeding innovative tasty vegetables and producing plant materials that could serve as source of multiple mogrosides.

In our study, the morphological observation results of transgenic tomato plants revealed that serious dwarfish was observed in the four independent tomato transgenic lines, but no obvious morphological changes were found in the transgenic cucumber plants. This result suggests that mogrosides can affect plant growth, which may produce metabolic toxic effects on transgenic tomato lines. The possible explanation for the absence of dwarfing in cucumber is that both cucumber and *Siraitia grosvenorii* are cucurbitaceae plants, and in cucumber itself there are many cucurbitane triterpene saponin such as cucurbitacin B and cucurbitacin C, and mogrosides are also triterpene saponin, in this case, there is no toxicity effect on the transgenic cucumber lines. Moreover, according to the previous researches, most studies have proven that transgenic engineering has caused a metabolic disturbance in gibberellin metabolism, which leads to the dwarfish in the transgenic tomato lines[52,53]. And the insertion of transgenic genes often causes poor growth, serious dwarf in transgenic plants, for example, previous study showed that the integration of T-DNA into the genome will activate or inactivate other nonspecific genes expression, which has caused physiological disorders or plant resistance in transgenic plants[54,55]. Therefore, we speculated that the dwarfish of transgenic tomato in this study may be correlated to the changes in gibberellin metabolism, but more information will be further introduced in several forthcoming experiments.

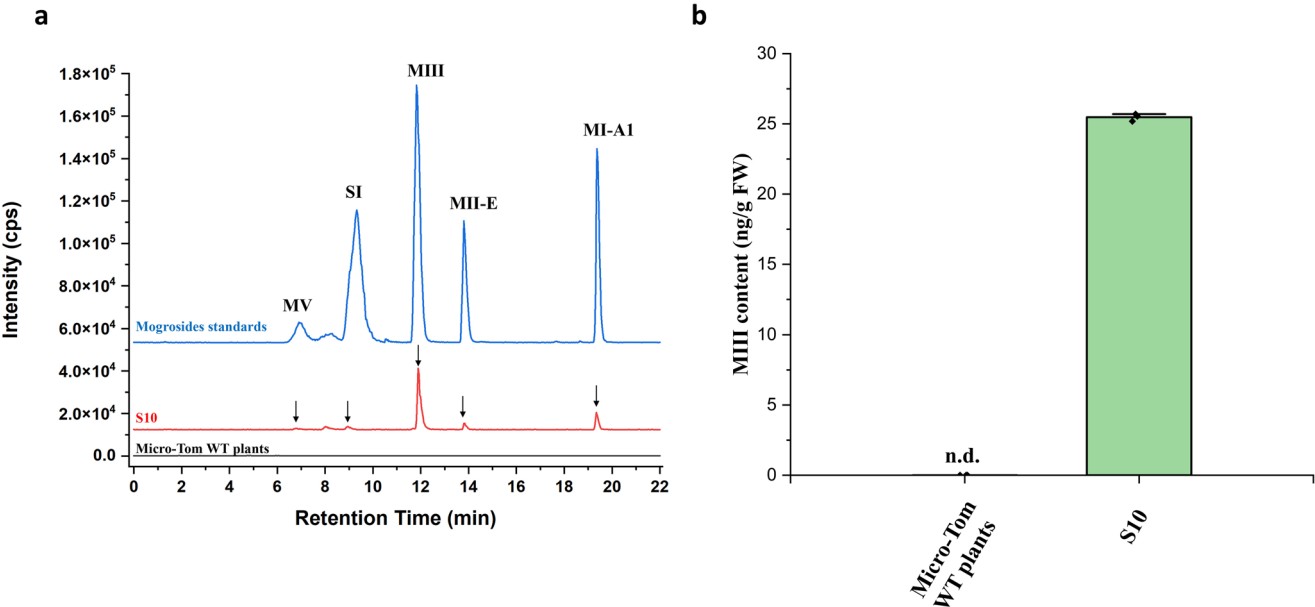

**Fig. 6 Production of mogrosides in tomato transgenic lines. a** HPLC-ESI-MS/MS analysis of mogrosides in transgenic tomato fruits. **b** Accumulation of MIII in transgenic tomato fruits. The black arrows indicate the peak of mogrosides. n.d., not detected. The data are presented as the mean values ± SDs, $n = 3$ biologically independent samples.

Multigene transformation can be used to introduce complete metabolic pathways into plants. Unlike the time-consuming and tedious steps of conventional cross breeding, re-transformation and co-transformation methods[56,57], emerging multigene vector transformation and polycistronic transgenes have excellent advantages. In particular, multiple vector transformation allows the assembly of multiple gene expression cassettes into a single T-DNA region and integration into the host chromosome genome[57,58]. To date, there is no evidence to suggest that the maximum number of transgenes can be introduced into plants, and yet little research has been conducted on 6 or more transgene transformations in a single vector. Additionally, as the number of transgenes rises, unstable binary vectors can induce spontaneous gene loss in heterologous plants[59,60]. Consequently, the schematic design of the multigene vector assembly in this research was important. The Transgene Stacking II system (TGS II)[61,62], Gateway Recombination system[63,64], Gibson Assembly[65,66] and In-fusion technology[67] have been widely used in multigene vector assembly. Among these, the Gateway recombination system remains a challenge with respect to the assembly of more than five genes due to operational difficulty. On this basis, TGS II was recently developed as a more convenient and efficient multigene vector system that can be used to transform 4–8 genes into heterologous plants[68]. Gibson assembly and In-fusion technology have been widely used to fuse multiple overlapping DNA fragments simultaneously in a single reaction. In addition, the maximum vector size of In-fusion technology is 46 kb. Therefore, In-fusion technology allows the assembly of multiple gene expression cassettes. Considering repetitive sequences in a multigene vector, self-cleaving 2A peptides (16 to 20 amino acids) was introduced, which lead to relatively high levels of downstream protein expression compared to other strategies for multi-gene expression[69–71]. However, the efficiency of protein expression mediated by different 2A sequences in this study, *SgSQE1, SgCS, SgEPH2, SgP450, SgUGT269-1* and *SgUGT289-3* transcript levels were different in the transgenic cucumber leaves and fruits. One possible reason is that most of transgenes driven by a constitutive promoter, CaMV 35S promoter, which showed that marked variations in transcriptional activity depending on the tissue and organ type[72–76]. CaMV 35S promoter drove β-glucuronidase (GUS) activity was higher in birch roots and axillary buds than in other birch organs[75]. Green fluorescence protein (GFP) driven by CaMV 35S promoter had a higher activity in the tobacco leaf vascular tissues than other tissues[74]. And physiological conditions and abiotic stress also affected the target transgenes expression. Therefore, to ensure high and stable transgene activity, considerable attention should be given to the multigene vector construction and growth condition further.

In summary, an In-fusion based gene stacking strategy for transgene stacking was developed, 6 mogrosides biosynthesis genes were transferred into cucumber and tomato, and transgenic sweet cucumber containing mogroside V and slightly sweet tomato containing mogroside III. This study displays the great potential in the genetic improvement of fresh-eating sweet plants. Our work fundamentally changes the traditional pattern of sweet-taste breeding, and offers a excellent mode for the transformation of non-sugar and non-protein sweet components instead of increasing sugar contents in fresh vegetables. Moreover, this study provides a possibility for mogrosides heterologous biosynthesis.

## Methods

**Plant materials and strains**. *Cucumis sativus* (JinYan 4, ZY4) and *Lycopersicon esculentum* (Micro-Tom) were used for plant transformation. Transient expression experiments were carried out in *Cucumis sativus* ZY4 and tobacco. *Escherichia coli* strains DH5α and XL10-Gold (WeidiBio, Shanghai, China) were used in this experiment, and *Agrobacterium tumefaciens* strain GV3101 was used for transformation.

**Promoter activity assays**. Appropriate promoters are crucial for driving the expression of target genes in multigene vectors. Ubiquitin 10 AtUBQ10 and serine carboxypeptidase-like *AtSCPL30* (AtPD7, 456 bp) promoters were cloned from *Arabidopsis thaliana* using KOD One PCR Master Mix (TOYOBO CO., LTD., Japan). The PCR conditions were as follows: initial activation at 98 °C for 5 min, followed by 35 cycles at 98 °C for 10 s, annealing temperature −5 °C for 5 s, and 68 °C for 10 s/kb, and a final extension at 68 °C for 7 min. To construct AtUBQ10:β-glucuronidase (GUS) and AtPD7:GUS vectors, each of these promoters was fused to a pBI121 plasmid at *XbaI/Bam*HI restriction sites (Fig. 1b). The primers were listed in the Supplementary Table S2. The resultant recombinant plasmids were confirmed by sequencing. pBI121 driven by the 35S cauliflower mosaic virus (CaMV 35S) promoter was used in this study as a positive control. All the plasmids were transformed into *Agrobacterium tumefaciens* strain GV3101 and used for transient assays in *Cucumis sativus* and *Nicotiana benthamiana*.

**Multigene expression vector construction**. The cDNA of *Siraitia grosvenorii* fruits was used to amplify the coding sequences of *SgSQE1* (squalene epoxidase), *SgCS* (cucurbitadienol synthase), *SgEPH2* (epoxide hydrolases), *SgP450* (cytochrome P450 monooxygenase), *SgUGT269-1* and *SgUGT289-3* (UDP-glucosyltransferases)[29]. To initiate the expression of all genes, the CaMV 35S promoter was isolated from the pBI121 plasmid, and the AtUBQ10 and AtPD7 promoters were amplified from *Arabidopsis thaliana* using KOD One PCR master Mix. In terms of transcriptional terminators, nopaline synthase (NOS) terminators were obtained from pBI121, and mannopine synthase (MAS) terminators and heat-shock protein (HSP) 18.2 terminators were chemically synthesized by GEN-EWIZ (Suzhou, China).

In the first round of gene assembly, each of these mogrosides biosynthesis-related genes was amplified via Phanta Max Super-Fidelity DNA Polymerase (Vazyme Biotech Co., Ltd., Nanjing, China), and then *SgCS*, *SgEPH2*, *SgP450* and *SgUGT289-3* were ligated into the *BamH*I and *Sac*I sites of the pBI121 vector using a ClonExpress II One Step Cloning Kit (Vazyme Biotech Co., Ltd., Nanjing, China). *SgSQE1* fused with AtPD7 promoter and HSP terminator, and *SgUgt269-1* together with the AtUBQ10 promoter and MAS terminator was inserted into pBI121 at the *Hind*III and *EcoR*I sites using a ClonExpress MultiS One Step Cloning Kit (Vazyme Biotech Co., Ltd., Nanjing, China), respectively. PD7:SgSQE1:Thsp, 35S:SgCS:Tnos, 35S:SgEPH2:Tnos, 35S:SgP450:Tnos, 35S:SgUGT289-3:Tnos and UBQ10:SgUGT269-1:Tmas were generate. In the second round of gene assembly, each of the first gene expression cassettes containing the promoter, target gene and terminator was cloned via KOD One PCR Master Mix. PD7:SgSQE1:Thsp and 35S:SgCS:Tnos were inserted at the *EcoR*I/*Hind*III restriction sites of a pCAMBIA1300 binary plasmid via a ClonExpress MultiS One Step Cloning Kit. Similarly, UBQ:SgUGT269-1:Tmas and 35S::SgUGT289-3:Tnos were ligated into the same restriction sites of the pCAMBIA1300 plasmid, yielding a double-gene expression cassette. In the third round of gene assembly, the regions of PD7:SgSQE1:Thsp::35S:SgCS:Tnos and 35S:SgEPH2:Tnos were amplified by PCR and subcloned into the UBQ10:SgUGT269-1:Tmas::35S:SgUGT289-3:Tnos *EcoR*I/*Hind*III sites of a pCAMBIA1300 plasmid. Then, the sequences of UBQ10:SgUGT269-1:Tmas::35S:SgUGT289-3:Tnos and 35S:SgP450:Tnos were isolated and inserted into the pCAMBIA1300 plasmid, and triple-gene expression cassettes were constructed. Subsequently, in the final round of gene assembly, the 6.1-kb sequence of UBQ10:SgUGT269-1:Tmas::35S:SgUGT289-3:Tnos::35S:SgP450 and SgSQE1:Thsp::35S:SgCS:Tnos::35S:SgEPH2:Tnos (6.5-kb) were isolated from the triple-gene expression cassettes. For the final multigene expression vector, above-mentioned fragments were ligated with a 2A peptide linker from porcine teschovirus (amino acid sequence is GSGATNFSLLKQAGDVEENPGP) (P2A) via a ClonExpress Ultra One Step Cloning Kit (Vazyme Biotech Co., Ltd., Nanjing, China). This final vector was referred to as the U22p-SCE vector in this study (Supplementary Fig. S2a). All primers used for multi-gene vector construction are listed in the Supplementary Table S3.

**Transient expression**. To verify the multigene expression vectors and analyse the promoter activity, a transient expression assay was performed in *Cucumis sativus* ZY4 and tobacco. A multigene expression vector was transformed into *Agrobacterium tumefaciens* strain GV3101 through the freeze-thaw method. The transformed *Agrobacterium* strains were cultured at 28 °C in LB media supplemented with rifampicin and kanamycin until the $OD_{600}$ of the bacteria reached 0.6. All the cultures were centrifuged at $5000 \times g$ for 10 min, resuspended in buffer that included 10 mM 2-(N-morpholino) ethanesulfonic acid (MES), 10 mM $MgSO_4$ and 200 μM acetosyringone (AS), and then incubated for 2–4 h at room temperature. The suspension was infected via a needleless syringe into cotyledons of 8-day-old *Cucumis sativus* ZY4 plants and leaves of *Nicotiana benthamiana* (positive control). To ensure accuracy, the transient expression procedures were repeated independently ten times.

**GUS staining**. *Nicotiana benthamiana* was sampled at 48 h post-infiltration, and cotyledons of *Cucumis sativus* ZY4 were sampled at 1, 3, 5 and 7 d for GUS histochemical staining analyses according to a previous protocol[42]. Images were taken using a Canon EOS 80D. The images were used to determine the intensity of blue staining. This experiment was repeated independently ten times.

**Generation of engineered cucumber plants**. The seeds of *Cucumis sativus* line ZY4 were soaked in the 55 °C for 15 min, and then placed at room temperature for 2 h. The cucumber seeds were sterilized following Trulson et al. with minor modification[77]. After soaking, seeds were sterilized with 75% alcohol for 30 s, followed by 3% NaClO for 15 min, then washed 5 times with sterile water, and inoculated on the Murashige and Skoog (MS) under darkness for 3 d. In order to obtain genetically engineered cucumber, we used the cotyledons of 4-day-old *Cucumis sativus* (cucumber) as explants. The cotyledons were cut from the seedlings and removed the growing point. The explants were placed on the pre-cultivation media, MS media containing 1 mg/L 6-benzylaminopurine (6-BA), 0.5 mg/L abscisic acid (ABA) and 2 mg/L $AgNO_3$ for 1 day in the dark. Then, the explants were infected with *Agrobacterium* harbouring the multigene expression vector for 30 min. and co-cultivated on co-cultivation media (MS media with 1 mg/L 6-BA, 0.5 mg/L ABA, 2 mg/L $AgNO_3$ and 1.45 mg/L AS) for 2 day in the dark at 25 °C. After co-cultivations, the explants were cultivated in selective medium (MS media with 1 mg/L 6-BA, 0.5 mg/L ABA, 2 mg/L $AgNO_3$, 500 mg/L Cefotaxime sodium (Cef) and 5 or 8 mg/L hygromycin (Hyg) under 18 h light/6 h dark conditions).

Hyg (5 mg/L) was used to select the transformed plants. Calli grew on MS media supplemented with Hyg (8 mg/L). Generally, the adventitious shoots that arose from calli reached were regenerated into green and healthy explants. Then, 2–3 cm segments of the regenerants were selected for cultivation in rooting media supplemented with 10 mg/L Hyg. After 1 month of cultivation, the regenerated cucumber seedlings were excised in vivo and then cultivated in a greenhouse (Fig. 2a). After that, the callus and regeneration plants were obtained by subculture continuously. To obtain transgenic lines, The regeneration plants were cultured on rooting media (MS media with 1 mg/L Gibberellin A3 ($GA_3$), 400 mg/L Cef). The regenerated plants were cultured in a greenhouse at 28 °C under 18 h /6 h (light/dark).

**Generation of transgenic Micro-Tom tomato plants**. The genetic transformation was performed following Sun et al. with modification[78]. Seeds of Micro-Tom were sterilized in 70% ethanol for 30 s and in 10% NaClO for 10 min, rinsed four times with sterilized water and dried on sterilized filter paper. The seeds were germinated in the MS media. Using the leaves, cotyledons and hypocotyls of 10-day-old Micro-Tom plants as explants, leaves and cotyledons were cut into the pieces about 0.5 cm × 0.5 cm, the hypocotyls were cut into segments about 0.5-0.6 cm to allow them to adsorb the bacterial suspension. They were pre-cultured on the pre-cultivation media (MS media with 1 mg/L 6-BA and 0.2 mg/L NAA) for 24 h. After pre-cultivation, explants were incubated for 30 min with *Agrobacterium* harbouring the multigene expression vector and then transferred to co-cultivation media (MS media containing 1 mg/L 6-BA, 0.2 mg/L NAA, and 0.1 mM AS) for 2 days. Then, the explants were transferred to selection media (2.0 mg/L zeatin (ZT), 0.2 mg/L indole-acetic acid (IAA), 10 mg/L Hyg and 500 mg/L carbenicillin). Afterwards, the buds were transferred to elongation media (1.0 mg/L ZT, 0.05 mg/L IAA, 500 mg/L Cef, 10 mg/L Hyg and 500 mg/L carbenicillin), and the regenerated shoots were allowed to grow to a length of 1 cm. The shoots were subsequently transferred to rooting culture media consisting of 1/2-strength MS media with 0.1 mg/L IAA, 2 mg/L Hyg, 500 mg/L Cef and 500 mg/L carbenicillin.

**Molecular identification**. Genomic DNA was obtained from the transgenic lines using a Plant Genomic DNA Kit (Tiangen Biotech Co., Ltd., Beijing). Genomic PCR with KOD One PCR Master Mix was performed to verify the transgenic plants. In addition, genomic PCR was used to detect the transgenic lines obtained by Hyg resistance screening. Wild-type (WT) plant genomic DNA was used as a negative control. All the primers used for PCR detection are shown in Supplementary Table S4.

Total RNA was isolated from the leaves of the transgenic plants via a CWBIO RNA extraction kit (CWBIO. Co., Ltd., Beijing, China), and 1 μg of total RNA was used to reverse transcribe cDNA via *TransScript*® One-Step gDNA Removal and cDNA Synthesis SuperMix (Transgen, Beijing, China). For quantitative real-time PCR (qRT-PCR), *PerfectStart*$^{TM}$ Green qPCR SuperMix (Transgen, Beijing, China) in conjunction with an ABI CFX96$^{TM}$ Real-Time System (USA) was used to quantify the expression levels according to the manufacturers' instructions. The thermal cycling was as follow: (1) 95 °C for 30 s; (2) 40 cycles of 3 s denaturation at 95 °C, 10 s annealing at 55 °C; (3) dissociation curve consisting of 15 s incubation at 95 °C, 60 s incubation at 60 °C, a ramp up to 95 °C. The gene transcript levels were precisely quantified for target gene expression with the $2^{-\Delta\Delta CT}$ method, and *Csactin* and *Leactin* were used as internal control. This experiment was conducted for multiple technical replicates. All the primers used for qRT-PCR are listed in Supplementary Table S5.

**Quantitative analysis of metabolites by HPLC-ESI-MS/MS**. All samples were ground in liquid nitrogen (cucumber, 10 g and tomato, 4 g), homogenized in 10 mL and 4 mL 80% methanol solution, respectively. And then an ultrasonic water-bath assisted extraction was performed at room temperature at 40 kHz for 1 h, and centrifuged at $5000 \times g$ for 20 min. Afterwards, the supernatant was collected and filtered using a 0.22 μM Millipore filter.

For quantitative analysis of mogrosides and mogrol contents, an AB SCIEX QTRAP 4500 LC-MS/MS (AB SCIEX, Toronto, Canada) system and an Agilent Technologies 1260 Series LC system (Agilent, USA) equipped with an Agilent Poroshell 120 SB C18 column (100 mm × 2.1 mm, 2.7 μm) were used. The mobile phase consisted of (A) water (including 0.1% formic acid) and (B) acetonitrile (including 0.1% formic acid) with a gradient elution. The HPLC conditions for mogrosides were as follows: 0 min, 20% B; 3–5 min, 23% B; 18 min, 40% B; and 18.01–20.10 min, 20% B. The flow rate was 0.25 mL/min. For mogrol, the HPLC conditions were as follows: 0 min, 20% B; 0.5 min 30% B; 2–4 min, 88% B; and 5.50–8.00 min, 20% B. The HPLC-ESI-MS/MS parameters are listed in Table 1, and electrospray ionization (ESI) with multiple reaction monitoring (MRM) scanning was used in this study. Each experiment was performed three times. The above two methods have been developed for quantitative analysis in our previous work, and they are suitable for the accurate quantification of target components in this study. The quantitative analyses were performed by means of an external standard method[79,80].

All standards including mogroside I-A1 (MI-A1), mogroside II-E (MII-E), mogroside III (MIII), siamenoside I (SI), mogroside V (MV) and mogrol were purchased from Chengdu Must Bio-Technology Co., Ltd. (Sichuan, China) and solved into the methanol.

**Table 1 HPLC-ESI-MS/MS parameters.**

| Analytes | Molecular Formula | Retention time (min) | product ion (m/z) | DP (V) | CE (eV) |
|---|---|---|---|---|---|
| MV | $C_{60}H_{102}O_{29}$ | 7.38 | 1285.8/1123.7 <br> 1285.8/961.7 | −220 | −90 |
| SI | $C_{54}H_{92}O_{24}$ | 9.62 | 1123.6/961.6 <br> 1123.6/799.2 | −220 | −75 |
| MIII | $C_{48}H_{82}O_{19}$ | 11.86 | 961.6/799.4 <br> 961.6/637.3 | −170 | −70 |
| MII-E | $C_{42}H_{72}O_{14}$ | 13.82 | 799.5/637.5 <br> 799.5/475.5 | −170 | −65 |
| MI-A | $C_{42}H_{72}O_{14}$ | 19.38 | 637.5/475.5 <br> 637.5/160.5 | −160 | −540 |
| Mogrol | $C_{30}H_{52}O_{4}$ | 4.01 | 459.3/441.2 <br> 459.3/423.3 | 80 | 20 |

| MS parameters | Mogrosides | Mogrol |
|---|---|---|
| Ion mode | Negative | Positive |
| Source temperature (°C) | 550 | 550 |
| Ionization voltage (V) | −4500 | 5500 |
| GS1 (psi) | 55 | 60 |
| GS2 (psi) | 55 | 50 |
| CUR (psi) | 20 | 20 |
| CAD | Medium | Medium |
| Dwell time (ms) | 100 | 200 |
| EP (V) | −10 | 10 |
| CXP (V) | −15 | 10 |

**Qualitative analysis of metabolites using UPLC-ESI-QTOF-MS/MS.** To identify metabolites in transgenic plants were analyzed by a UPLC-ESI-QTOF-MS/MS system (Waters Corp.) with negative electrospray ionization (ESI). The MS conditions were in MSE continuum mode, with a scan range of m/z 100 to 1500. The collision voltage was 2.5 KV. The sampling cone voltage and extractor voltage were 40 KV and 4 KV, respectively. The source temperature and desolvation temperature were 100 °C and 250 °C, respectively. The cone gas flow was 50 L/hr, and desolvation gas flow was 600 L/hr; Collision energy was ramped from 15 to 45 eV for the collection of MS/MS data. Masslynx 4.1 software was used for data acquisition and processing.

**Statistics and reproducibility.** The real-time quantitative analysis data are presented as the means ± SEM of at least three independent experiments in the paper. Unless otherwise stated, all samples were obtained randomly and all independent samples or experiment were repeated more than three times. The average values and standard deviations were calculated using SPSS 16.0 statistics programme (IBM Co., Armonk, NewYork, USA). All plots were obtained using Origin 2019b (OriginLab Co., Northampton, MA, USA).

**Reporting summary.** Further information on research design is available in the Nature Portfolio Reporting Summary linked to this article.

## Data availability
Unprocessed blot images are shown in Supplementary Fig. S7. Source data for the figures is available in Supplementary data 1. This plasmid Up-SCE is available through Addgene Catalogue # 197269. All other relevant data are available from the corresponding author upon reasonable request.

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

## Acknowledgements

This work was financially supported by the National Natural Science Foundation of China (U20A2004; 81973413; 82104548), the CAMS Innovation Fund for Medical Sciences (CIFMS) (2019-1007-15; 2021-I2M-1-071), and the Science and Technology Major Project of Guangxi (GuiKeAA19254025).

## Author contributions

J.L., X.M. and J.Q conceived this study. J.L., T.L. and L.X. constructed multigene vector. J.L., C.M. and X.H. performed the plant transformation assay. S.C., J.L. and X.J. performed the PCR detection and qRT-PCR analysis. Z.L. detected the mogrosides components in the transgenic plants. X.M. supervised the project. J.L. drafted the manuscript. X.M., Z.L. and T.L. revised the manuscript.

## Competing interests

The authors declare no competing interests.
