## [Peer Review File · Communications Biology]

Reviewers' comments:

Reviewer #1 (Remarks to the Author):

Liao et al. reported the expression of some glucosylated mogrol in tomato and cucumber plants, for the first time, using transgenic technology. Some issues regarding the analysis of the aglycone (mogrol) and the conclusion regarding the flavor needs to be addressed.

Line 85: according to previous published works, mogroside III is also very bitter compound.

Line 85-86: this line sentence needs to be referenced.

Line 248-279: the authors need to specify, in detail, how the quantitation was carried out. Did the author used any internal standard?

Line 392-393: according to the methods, the authors used a C18 column, could the authors explain why the mogrol has a lower retention time than the glucosylated analogs? see also comment below.

Line 394-395: the figure 3 does not show the mogrol content, neither the chromatogram. How did the authors quantified the mogrol included in the figure S4?

Line 464 -466: according to this phrase, the authors carried out a tasting test. Please, indicate in the method how the authors carry out this analysis. This assumption is also made in the line 548. Even if there is mogroside V, in order to determine if the tomato is sweet, the authors need to address the flavor analysis.

Reviewer #2 (Remarks to the Author):

GENERAL COMMENTS

I would like to congrats the authors on this complete work. Frankly, I think is very well written and contrasted with the literature. I also think that the information provide is exquisite. There are many different techniques utilised from chemical characterization to biomolecular techniques, very interesting. However, I would like to provide some suggestions to improve as far as possible this paper and I hope my corrections are helpful. I am very pleased to have been called to review this article since also made me learn a lot of concepts and made me grow professionally.

To start, I think it should be better stated the need to improve vegetable flavours to justify better the need for them to genetically produce mogrosides. On the contrary, I agree that is a new synthesis pathway of mogrosides.

TITLE

- In my opinion, the title is not so revealing, at least for the part where it is said "a sweet flavour and a new mode". When you read this title you are not so sure what it is talking about, a new mode of what? This could reduce the impact of your article in such an important journal.

ABSTRACT

- In the first sentence of the abstract (lines 39-40), needs to be homogenized verbal tenses since does not make sense "which were approved, and which has a broad of applications". I would suggest correcting for "which have had a broad application over the past decades"

INTRODUCTION

I really like the introduction, it leads you perfectly to introduce the context of the manuscript.

- I would include some regulatory status of mogrosides when the introduction starts presenting mogrosides like in the abstract where it was mentioned the approval of mogrosides by FDA in 2010.

MATERIALS AND METHODS

-Section Generation of transgenic Micro-Tom tomato plants: I am not convinced about using the symbol + as the description of the components of the media, would not be better to choose another more suitable way to indicate what is the "recipe" based on? Like only using commas.

-Section HPLC-MS/MS-based analysis of mogrosides: "ultrasonically extracted" please specify the material in this section, is it a bath ultrasound? Is it a probe ultrasound?

- I missed more information about the PCR, in what consisted of the thermal cycling program of the

PCR amplification?

-Section Data analysis: SPSS was used for the statistics, please complete this section with the statistical approach used. To reveal the significant differences by ANOVA? Was a multivariate or a single-variate analysis performed? What was done exactly in this program?

RESULTS

-It can be obvious that non-transgenic tomato or cucumber do not present mogrosides compounds but since the non-transgenic group would be the controls, this could have been compared at some point to highlight that the biosynthesis carried out. In Figure 3: even if the LC-MS/MS chromatograms are properly compared with mogroside standards, this qualitative data, to me do not match with the quantitative data. If I am understanding good, regarding the U1 fruit profile in Figure 3A, there is barely MV, however in Figure 3B, the bars indicate around 600 ng/g FW of MV as for MIII. MIII has not the same peak abundance as MV in the chromatograms.

-Where is the purpose of using SPSS if there are not even significant differences between the results? The statistics were only focused on the standard deviations?

-Figure 4: I think the figure legend can bring confusion. Numbers 1, 2, 3, 4, 5 represented the mogroside V, siamenoside I, mogroside III, and so on? It confuses me since the number of the nomenclature of the mogrosides is based on the glucose number attached to the mogrol aglycone. So here I understand that number 1 is mogroside V? number 2 siamenoside I? etc? Also, the red arrows get me lost, why there is not in number 3? Or which one it is? There are two peaks eluting at close retention times.

- Figure 5c: please improve the quality of the graphs, it is difficult to read the values and titles. In my opinion, I insist that the statistics are lacking in this manuscript, it is made a very basic statistical analysis to mention SPSS.

DISCUSSION

I think is not mentioned the (for sure significant) quantity of mogroside III in the cucumber line.

-It is the first time to have biosynthesized mogrosides in vegetables? If it is the case you should highlight even more in the abstract, discussion and conclusion.

-The concentration of mogrosides fluctuates during the ripening process, is this going to be affected during the cultivation of these vegetables?

-In introduction was commented that mogroside was chosen over different compounds of *Stevia rebaudiana*, glycyrrhizin... due to the taste. However, the number of glucose units included in the aglycone influences the taste perception including the bitter taste, it is described that the presence of fewer than three glucoses results in being tasteless or having a bitter taste. Since this was never mentioned, was that considered? Your purpose what to produce mogroside V and for example in tomato line the production culminated in mogroside III. And at least there are trace amounts of mogrosides I and II, how much this could affect the global sensorial analysis? Or this quantity is significantly lower than mogrosides III and V to be officially considered trace amounts?

All the best to the authors,

Signed by Ana Muñoz

Dear reviewers,

Thank you for your letter and for your comments concerning our manuscript entitled "A new model of breeding sweet vegetables: synthesizing diverse mogrosides in cucumber and tomato by genetic manipulation", which submitted to Communications Biology. Those comments are all valuable and very helpful for revising and improving our paper, as well as the important guiding significance to our researches. We have studied comments carefully and have made correction which we hope meet with approval. The main corrections in the paper and the responds to the reviewer's comments are as following:

Reviewer #1:

Response to comment:

1. Line 85: according to previous published works, mogroside III is also very bitter compound.

Reply: In fact, it is reported in literature that mogroside III is 195 times sweeter than sucrose (Murata, Y., Yoshikawa, S., Suzuki, Y. A., Sugiura, M., Inui, H., Nakano, Y., Sweetness characteristics of the triterpene glycosides in *Siraitia grosvenori*. J. Jpn. Soc. Food Sci. Technol. 2006, 53, 527–533.). Its taste description is weak or tasteless compared with mogroside V, and does not have bitterness, bitterness are lower monoglycosides and diglycosides.

2. Line 85-86: this line sentence needs to be referenced.

Reply: Thanks for your comments. We have added the reference in this line.

3. Line 248-279: the authors need to specify, in detail, how the quantitation was carried out. Did the author used any internal standard?

Reply: Two quantitative analysis methods for mogrol and mogrosides have been developed in our previous work (Luo Z, Zhang K, Shi H, et al. Development and Validation of a Sensitive LC–MS-MS Method for Quantification of Mogrol in Rat Plasma and Application to Pharmacokinetic Study. Journal of Chromatographic Science, 2017(3): 284-290; Jing Q, Luo Z, Zhe G, et al. Identification of a Novel Specific Cucurbitadienol Synthase Allele in *Siraitia grosvenorii* Correlates with High Catalytic Efficiency. Molecules, 2019, 24(3):627). These articles have been referred in the manuscript. The quantitative analyses were performed by means of an external standard method.

4. Line 392-393: according to the methods, the authors used a C18 column, could the authors explain why the mogrol has a lower retention time than the glucosylated analogs? see also comment below.

Reply: Due to the mogrosides in negative ion mode can obtain the ionization effect that

satisfies the development of quantitative methods, while it is difficult to find corresponding characteristic ions for mogrol in negative mode. Finally, we established two quantitative methods for mogrol and mogrosides, respectively. The mogrol has a lower retention time because the quantitative method for mogrol uses a stronger elution procedure.

5. Line 394-395: the figure 3 does not show the mogrol content, neither the chromatogram. How did the authors quantified the mogrol included in the figure S4?

Reply: The quantitative analysis methods for mogrol refer to our previous work (Luo Z, Zhang K, Shi H, et al. Development and Validation of a Sensitive LC–MS-MS Method for Quantification of Mogrol in Rat Plasma and Application to Pharmacokinetic Study. *Journal of Chromatographic Science*, 2017(3):284-290). The mogrol content and the chromatogram have been added in the Supplementary Fig S4a-d.

6. Line 464 -466: according to this phrase, the authors carried out a tasting test. Please, indicate in the method how the authors carry out this analysis. This assumption is also made in the line 548. Even if there is mogroside V, in order to determine if the tomato is sweet, the authors need to address the flavor analysis.

Reply: The samples are precious, so we have not performed the flavour analysis, the taste is presumed according to the sweetness of mogrosides and natural vegetable flavours. To determine the taste perception of transgenic vegetables, we planned to do more genetic engineering experiments in the future.

Reviewer #2:

Response to comment:

1. TITTLE

- In my opinion, the title is not so revealing, at least for the part where it is said “a sweet flavour and a new mode”. When you read this title you are not so sure what it is talking about, a new mode of what? This could reduce the impact of your article in such an important journal.

Reply: Thanks for your comments and they are very valuable and helpful for revising and improving our paper. After our discussion, we thought that the title “A new model of breeding sweet vegetables: synthesizing diverse mogrosides in cucumber and tomato by genetic manipulation” could be an option.

2. ABSTRACT

- In the first sentence of the abstract (lines 39-40), needs to be homogenized verbal tenses since does not make sense “which were approved, and which has a broad of applications”. I would suggest correcting for “which have had a broad application over the past decades”

Reply: Thanks for your comments and we have modified this sentence.

3. INTRODUCTION

- I would include some regulatory status of mogrosides when the introduction starts presenting mogrosides like in the abstract where it was mentioned the approval of mogrosides by FDA in 2010.

Reply: Thanks for your comments. some regulatory status of mogrosides have been introduced in text.

4. MATERIALS AND METHODS

(1)-Section Generation of transgenic Micro-Tom tomato plants: I am not convinced about using the symbol + as the description of the components of the media, would not be better to choose another more suitable way to indicate what is the “recipe” based on? Like only using commas.

Reply: Thanks for your comments. We have been modified the description of components of the media using comma.

(2)-Section HPLC-MS/MS-based analysis of mogrosides: “ultrasonically extracted” please specify the material in this section, is it a bath ultrasound? Is it a probe ultrasound?

Reply: Thanks for your comments. an ultrasonic water-bath assisted extraction was used and we have changed in text.

(3)- I missed more information about the PCR, in what consisted of the thermal cycling program of the PCR amplification?

Reply: Thanks for your comments. the thermal cycling program of the PCR amplification has been added in text.

(4)-Section Data analysis: SPSS was used for the statistics, please complete this section with the statistical approach used. To reveal the significant differences by ANOVA? Was a multivariate or a single-variate analysis performed? What was done exactly in this program?

Reply: The average values and standard deviations were calculated using SPSS 16.0 statistics program. As we all known, non-transgenic tomato or cucumber do not present mogrosides compounds and the mogroside biosynthesis genes are not identified in the non-transgenic tomato or cucumber, in this case, we have not added the significant differences in the plot. Because we have found all data of transgenic lines group showed extremely significantly difference with the non-transgenic tomato or cucumber group.

5. RESULTS

(1)-It can be obvious that non-transgenic tomato or cucumber do not present mogrosides compounds but since the non-transgenic group would be the controls, this could have been compared at some point to highlight that the biosynthesis carried out. In Figure 3: even if the LC-MS/MS chromatograms are properly compared with mogroside standards, this qualitative data, to me do not match with the quantitative data. If I am understanding good, regarding the U1 fruit profile in Figure 3A, there is barely

MV, however in Figure 3B, the bars indicate around 600 ng/g FW of MV as for MIII. MIII has not the same peak abundance as MV in the chromatograms.

Reply: In this study, the quantitative analysis of mogrosides was performed under MRM mode by HPLC-ESI-MS/MS. The quantitative analyses were performed by means of an external standard method. In fact, for the target compounds of this study, peak abundance does not directly reflect the concentration of the compounds. Because at the same concentration (In figure 3a, the concentration of mogroside standards MV and MIII are both 200 ng/mL), the MV and MIII peak abundance are quite different, the peak abundance of MV is much less than MIII.

(2)-Where is the purpose of using SPSS if there are not even significant differences between the results? The statistics were only focused on the standard deviations?

Reply: The average values and standard deviations were calculated by SPSS 16.0 statistics program. Non-transgenic tomato or cucumber do not present mogrosides compounds and the mogroside biosynthesis genes, in this case, we have not added the significant differences in the plot. Because we have found all data of transgenic lines group showed extremely significantly difference with the non-transgenic tomato or cucumber group. And our purpose is proving the existence of mogrosides component or the mogrosides genes expression, in this case, we have not added the significant differences in the plot.

(3)-Figure 4: I think the figure legend can bring confusion. Numbers 1, 2, 3, 4, 5 represented the mogroside V, siamenoside I, mogroside III, and so on? It confuses me since the number of the nomenclature of the mogrosides is based on the glucose number attached to the mogrol aglycone. So here I understand that number 1 is mogroside V? number 2 siamenoside I? etc? Also, the red arrows get me lost, why there is not in number 3? Or which one it is? There are two peaks eluting at close retention times.

Reply: Thanks for your comments. Numbers 1, 2, 3, 4, 5 have been changed as MIA-1, MII-E, MIII, SI, and MV in Figure 4. The red arrows are used to identify the target mogrosides in the sample.

(4)- Figure 5c: please improve the quality of the graphs, it is difficult to read the values and titles. In my opinion, I insist that the statistics are lacking in this manuscript, it is made a very basic statistical analysis to mention SPSS.

Reply: Thanks for your comments. We have changed the Figure 5 with high quality. We analysed the average values and standard deviations using SPSS 16.0 statistics program. We have referred that non-transgenic tomato or cucumber do not present mogrosides compounds and the mogroside biosynthesis genes are not identified in the non-transgenic tomato or cucumber, and all data of transgenic lines group showed extremely significantly difference with the non-transgenic tomato or cucumber group. And our purpose is proving the existence of mogrosides component or the mogrosides genes expression, in this case, we have not added the significant differences in the plot.

6. DISCUSSION

(1)I think is not mentioned the (for sure significant) quantity of mogroside III in the cucumber line.

Reply: Thanks for your comments. The quantity of mogroside III have been added in the text.

(2)-It is the first time to have biosynthesized mogrosides in vegetables? If it is the case you should highlight even more in the abstract, discussion and conclusion.

Reply: Thanks for your comments. We have highlighted the production of mogrosides in vegetables firstly.

(3)-The concentration of mogrosides fluctuates during the ripening process, is this going to be affected during the cultivation of these vegetables?

Reply: The transgenic lines were weak and they were attacked by pests, which probably due to the sweet flavour. In the case, the samples were insufficient. And we only analysed the concentration of mogrosides in two different periods, and it wasn't significantly different. It probably due to the sampling period, we need to improve the situation in the future.

(4)-In introduction was commented that mogroside was chosen over different compounds of *Stevia rebaudiana*, glycyrrhizin... due to the taste. However, the number of glucose units included in the aglycone influences the taste perception including the bitter taste, it is described that the presence of fewer than three glucoses results in being tasteless or having a bitter taste. Since this was never mentioned, was that considered? Your purpose what to produce mogroside V and for example in tomato line the production culminated in mogroside III. And at least there are trace amounts of mogrosides I and II, how much this could affect the global sensorial analysis? Or this quantity is significantly lower than mogrosides III and V to be officially considered trace amounts?

Reply: In fact, it is reported in literature that mogroside III is 195 times sweeter than sucrose (Murata, Y., Yoshikawa, S., Suzuki, Y. A., Sugiura, M., Inui, H., Nakano, Y., Sweetness characteristics of the triterpene glycosides in *Siraitia grosvenori*. *J. Jpn. Soc. Food Sci. Technol.* 2006, 53, 527–533.). Its taste description is weak or tasteless compared with mogroside V, and does not have bitterness, bitterness are lower monoglycosides and diglycosides. We have added this content in the manuscript. To be honest, the initial goal is the transgenic tomato with mogroside V, however, we have not identified the mogroside V in the transgenic tomato lines. We think that this is an extremely complicated reason, and further study is needed. Although it is well known that mogroside IA-1, II-E are bitter-tasting glycosides, SI and MV are extremely sweet, MIII is tasteless or slightly sweet, which, which cultivate cucumbers to taste sweet, blend flavors, or totally new tastes to commonly known vegetables. And the content of MI-A and MII-E with bitter flavor were present at too low a level in the fruits of transgenic cucumber, and therefore the consequence for flavor of transgenic cucumber is relatively small. Although it seems that the quantity of mogrosides III and V is trace, their sweetness is extremely high. This has very little impact on the flavor of transgenic

lines.

All the best,

Xiaojun Ma

Reviewers' comments:

Reviewer #1 (Remarks to the Author):

I went through the manuscript. The authors answered the raised questions and comments by the two reviewers.

Reviewer #2 (Remarks to the Author):

Thank you for carefully answering and considering each of my notes, I appreciate it.

Reviewer #3

The authors introduced six mogrosides biosynthesis genes isolated from *Siraitia grosvenorii* into important vegetables such as cucumber and tomato, resulting in succeeded in creating mogrosides-accumulating plants by metabolic engineering strategies. I think that the topic may be of interest to the field. The reviewer's comments are as follows.

[Major comments]

1. For the transgenic tomato, it requires a more in-depth investigation. In particular, data in Figure 6 do not give a complete picture of the transgenic tomato lines. Data like Figure 3b and 4 should be shown to show how much other mogrosides such as MI-A1 and MII-E were accumulated in the tomato lines. Otherwise, we cannot fully understand what happened in the tomato lines.
2. In the latter part of the discussion section, the authors described transgene stacking of vectors (Lines 537–576), but this should be more compact. The reviewer think that the really important point is to discuss the phenotype of the transgenic lines. In particular, why SI and MV were not accumulated in the tomato lines even though six genes were expressed at "high transcript levels" should be discussed based on the characteristics of enzymes isolated from *Siraitia grosvenorii* (Line 511–515). Additionally, why did the growth stunt in the tomato transgenic line while the cucumber U1 line grew as wild type?

[Specific comments]

3. Line 109–114: For the historical perspective, it would be good to describe Golden Rice as well.
4. "PBI121," which can be found in Lines 139, 150 and Supplementary Figure S2, should be written as "pBI121."
5. Line 207–233: The authors do not provide sufficient information in the generation of transgenic plant lines and readers cannot reproduce the same procedures the authors did. Please describe how the experiment was conducted, what was the growth conditions of the culture, etc., citing appropriate references.
6. Line 208: "leaves" should be written as "cotyledons."
7. Line 211: Ref 41 does not properly describe the transformation method. Please cite an appropriate reference.
8. Line 214: "infiltrated" should be written as "infected," if the author did not vacuum during *Agrobacterium* infection.
9. Line 215, 217: MS media containing phytohormones, etc. cannot be indicated as "MS media." Change "MS media" to an appropriate name, such as "co-cultivation media," "shoot induction media," etc.
10. Line 218: Please indicate the composition of the root induction media.
11. Line 223: How old were the tomato plants used for the preparation of the explants?
12. Line 225: Show the conditions of pre-cultivation, citing the appropriate references of Micro-tom transformation.

13. Line 359: the word "sweet." I think it might be acceptable to use the word "sweet cucumber" in introduction or discussion sections. However, it would not be better to use this word in the Results section. Change to a scientifically correct title, such as "Generation of mogroside-accumulating cucumber." I don't feel I can fit with using the term "sweet cucumber" even though no one has eaten it.
14. Line 363–370: These sentences correspond to the method section.
15. 16. Line 370–371: The authors need to see the real data for the growth data in U1 line and WT.
16. Line 379: What does U1 stand for?
17. Line 380–382: A complete picture of the transformation experiment should be given. A table should be presented showing the total number of explants, the number of Hyg-resistant lines, and the number of lines introduced truncated T-DNA regions, etc.
18. Line 392: The expression levels of SgCS, for example, differ more than 30-fold between fruits and leaves, and it seems unreasonable to describe them as "similar". Please describe the difference in the expression levels of each gene accurately and in detail.
19. Line 446–448: A table should be presented showing the total number of explants, the number of Hyg-resistant lines, and the number of lines introduced truncated T-DNA regions (same as Line 380-382).
20. Line 454: "expressed" should be written as "introduced" or "inserted"?
21. Line 481–484: These sentences correspond to the discussion section.
22. Line 498: The contents of mogrosides per DW is given, but Figure 3b shows the amount of mogrol per FW. Is there any point in showing the contents of mogrosides per DW here? If not, it would be better to use the amount per FW in accordance with Figure 3b, so that readers would not be confused.
23. Line 528–529: "Our study is interesting" The reviewer really think the study is interesting. However, it would be better to revise this sentence.
24. Line 543–545: Please cite some references to past successes, for example, Ogo et al. *Plant Biotechnol J*, 11:734-746 (2013), and Zhu et al. *Mol Plant*, 10:918-929 (2017) as in ref 58.
25. The legend in Figure 1a and 1b could be reversed.
26. Figure 1c, it should be stated that GUS staining was performed after 48 h of infiltration in the legend.
27. Figure 2a, describes what each picture represents in each legend.
28. Figure 2b, add scale bars.
29. Figure 2c, each figure is too small to distinguish. They should be enlarged with a molecular marker.
30. Figure 5a should include a photo of a WT tomato that grew up at the same time.
31. Figure 5b, pointed in Figure 2c, the figure should be enlarged to clearly show the size of the amplified bands and which sample is in each lane.
32. Figure 5c, the bar graph in the upper right corner, the description of the vertical axis is missing. "Relative expression of SgEPH2"?

Editor's Comment: Authors only obtained a single independent transgenic line of the sweet cucumber (Lines 376-377). Results based on only a single independent transgenic line are almost never acceptable.

Dear reviewers,

We feel great thanks for your professional review work on our manuscript entitled "A new model of breeding sweet vegetables: synthesizing diverse mogrosides in cucumber and tomato by genetic manipulation". As you are concerned, there are several problems that need to be addressed. According to your nice suggestions, we have made extensive corrections to our previous draft, the detailed corrections are listed below.

Reviewer #3:

Response to comment:

[Major comments]

1. For the transgenic tomato, it requires a more in-depth investigation. In particular, data in Figure 6 do not give a complete picture of the transgenic tomato lines. Data like Figure 3b and 4 should be shown to show how much other mogrosides such as MI-A1 and MII-E were accumulated in the tomato lines. Otherwise, we cannot fully understand what happened in the tomato lines.

Reply: Thank you for your suggestion. According to the previous measured result, the content of MIII was relatively lower in the transgenic tomato, and we did not detect other mogrosides. In this case, we believed that the content was too low in the transgenic plants. Therefore, to enrich the mogrosides in the transgenic tomato, previous measured samples were concentrated 15 times. After that, the content of mogrosides were analyzed again, and this result was consistent with previous result. The content of MIII was 25.92 ng/g, and we also found a small amount of MI-A1 (5.65 ng/g), MII-E (2.33 ng/g), SI and MV. However, the content of SI and MV is still below the limit of quantification (<LOQ) (shown in modified Figure 6a), And the target compounds in tomato transgenic plants were further confirmed by UPLC-ESI-QTOF-MS/MS. The result has been shown in Supplementary Figure S6. In general, tomato transgenic plants also produced mogrosides, but the level was lower compared to mogrosides content in cucumber transgenic plants. Over the next few years, our group

will focus on optimizing this transformation system and establish more high-efficient and stable regeneration and genetic transformation system with multi-gene vector to promote mogrosides accumulation.

2. In the latter part of the discussion section, the authors described transgene stacking of vectors (Lines 537–576), but this should be more compact. The reviewer think that the really important point is to discuss the phenotype of the transgenic lines. In particular, why SI and MV were not accumulated in the tomato lines even though six genes were expressed at "high transcript levels" should be discussed based on the characteristics of enzymes isolated from *Siraitia grosvenorii* (Line 511–515). Additionally, why did the growth stunt in the tomato transgenic line while the cucumber U1 line grew as wild type?

Reply: Thank you for your suggestion. Firstly, we have re-organized or compressed the description of transgene stacking of vector in Discussion. Secondly, the phenotype of the transgenic lines has been discussed. Thirdly, in our supplementary experiments, the synthesis of mogrosides in transgenic tomato has been further confirmed, but the content is still lower, which has been answered in question 1. Finally, according to the previous literatures, insertion of transgenic genes often causes poor growth, serious dwarf in transgenic plants, for example, previous study showed that the integration of T-DNA into the genome will activate or inactivate other nonspecific genes expression, which has caused physiological disorders or plant resistance in transgenic plants (Henikoff S et al., 2004; Muir SR et al., 2001). And in our study, the mechanism of slow-growing need to be studied further.

[Specific comments]

3. Line 109–114: For the historical perspective, it would be good to describe **Golden Rice** as well.

Reply: We have added the suggested reference to the manuscript on Ref 32.

4. "PBI121," which can be found in Lines 139, 150 and Supplementary Figure S2,

should be written as "pBI121."

Reply: Thank you for pointing this out. The reviewer is correct, and we have revised the PBI121 into pBI121 within the whole manuscript and supplementary information.

5. Line 207–233: The authors do not provide sufficient information in the generation of transgenic plant lines and readers cannot reproduce the same procedures the authors did. Please describe how the experiment was conducted, what was the growth conditions of the culture, etc., citing appropriate references.

Reply: We think this is an excellent suggestion. We have provided more specific information in the generation of transgenic plant lines where the change can be found in the revised manuscript. We sincerely appreciate the valuable comments. We have added other literature in this section in the revised manuscript.

6. Line 208: "leaves" should be written as "cotyledons."

Reply: We sincerely thank the reviewer for careful reading. As suggested by the reviewer, we have corrected the leave into cotyledons.

7. Line 211: Ref 41 does not properly describe the transformation method. Please cite an appropriate reference.

Reply: As suggested by the reviewer, we have cited more appropriate reference to support this idea (Ref 43).

8. Line 214: "infiltrated" should be written as "infected," if the author did not vacuum during *Agrobacterium* infection.

Reply: Thanks for your careful checks. Based on your comments, we have made the corrections to "infiltrated" with "infected."

9. Line 215, 217: MS media containing phytohormones, etc. cannot be indicated as "MS media." Change "MS media" to an appropriate name, such as "co-cultivation media," "shoot induction media," etc.

Reply: We have carefully checked the manuscript and corrected the errors accordingly. MS media containing phytohormones has been revised by an appropriate name in the revised manuscript.

10. Line 218: Please indicate the composition of the root induction media.

Reply: We have added the suggested content to the manuscript on the **Materials and methods**.

11. Line 223: How old were the tomato plants used for the preparation of the explants?

Reply: 10-day-old tomato plants were used for the preparation of the explants in this assay. And this information has been provided the generation of transgenic Micro-Tom tomato plant lines.

12. Line 225: Show the conditions of pre-cultivation, citing the appropriate references of Micro-tom transformation.

Reply: We sincerely appreciate the valuable comments. We have added the reference on the **Materials and methods** in the revised manuscript (Ref 43).

13. Line 359: the word "sweet." I think it might be acceptable to use the word "sweet cucumber" in introduction or discussion sections. However, it would not be better to use this word in the Results section. Change to a scientifically correct title, such as "Generation of mogroside-accumulating cucumber." I don't feel I can fit with using the term "sweet cucumber" even though no one has eaten it.

Reply: We think this is an excellent suggestion. We have replaced "sweet cucumber" with "mogrosides-accumulating cucumber." in result sections the word "sweet cucumber" in introduction or discussion sections in the revised manuscript.

14. Line 363–370: These sentences correspond to the method section.

Reply: We agree with the reviewer's assessment. Accordingly, these sentences have removed into the method section.

15. 16. Line 370–371: The authors need to see the real data for the growth data in U1 line and WT.

Reply: We agree with the reviewer’s assessment. Accordingly, we have revised this sentence in the revised manuscript.

16. Line 379: What does U1 stand for?

Reply: Thank you for the reviewer’s valuable feedback, U1 represented the transgenic cucumber line which have been wrote more clearly in the revised manuscript.

17. Line 380–382: A complete picture of the transformation experiment should be given. A table should be presented showing the total number of explants, the number of Hyg-resistant lines, and the number of lines introduced truncated T-DNA regions, etc.

Reply: Thank you for this suggestion. A complete picture of the transformation experiment has been shown in the Figure 2a. the total number of explants, the number of Hyg-resistant lines and the number of lines introduced truncated T-DNA regions have been added in the **Results of Generation of mogrosides-accumulating cucumber section** in the revised manuscript.

18. Line 392: The expression levels of SgCS, for example, differ more than 30-fold between fruits and leaves, and it seems unreasonable to describe them as "similar". Please describe the difference in the expression levels of each gene accurately and in detail.

Reply: Thank you for this suggestion. We have modified the “similar” with “varied”. In fact, the difference in the expression levels of transgene in the leave and fruits were affected by the complex of multi-gene vector, the same promoters and physiological conditions. Previous study indicated transgenes driven by a common constitutive promoter can exhibit marked variations in transcriptional activity depending on plant organ, physiological conditions and in response to abiotic stress (<https://doi.org/10.1007/s11033-021-06235-x>). And we have re-wrote this part, and

described the difference in the Discussion.

19. Line 446–448: A table should be presented showing the total number of explants, the number of Hyg-resistant lines, and the number of lines introduced truncated T-DNA regions (same as Line 380-382).

Reply: Thank you for this suggestion. A complete picture of the transformation experiment has been shown in the Figure 2a. the total number of explants, the number of Hyg-resistant lines and the number of lines introduced truncated T-DNA regions have been added in the **Results of Generation of MIII-accumulating tomato section** in the revised manuscript.

20. Line 454: "expressed" should be written as "introduced" or "inserted"?

Reply: Thank you for pointing this out. “expressed” has been corrected on “introduced”.

21. Line 481–484: These sentences correspond to the discussion section.

Reply: We agree with the reviewer’s assessment. Accordingly, these sentences have removed into the discussion section.

22. Line 498: The contents of mogrosides per DW is given, but Figure 3b shows the amount of mogrol per FW. Is there any point in showing the contents of mogrosides per DW here? If not, it would be better to use the amount per FW in accordance with Figure 3b, so that readers would not be confused.

Reply: We feel sorry for our carelessness. In our resubmitted manuscript, the error is revised. The contents of mogroside was accumulated using Fresh weight. We have revised the DW with FW within the whole manuscript. Thanks for your correction.

23. Line 528–529: "Our study is interesting" The reviewer really think the study is interesting. However, it would be better to revise this sentence.

Reply: Thank you for pointing this out. The reviewer is correct, and we have revised the sentence in the revised manuscript.

24. Line 543-545: Please cite some references to past successes, for example, Ogo et al. *Plant Biotechnol J*, 11:734-746 (2013), and Zhu et al. *Mol Plant*, 10:918-929 (2017) as in ref 58.

Reply: As suggested by the reviewer, we have added more references to support this idea (Ref 32 and 63).

25. The legend in Figure 1a and 1b could be reversed.

Reply: Thank you for pointing this out. The reviewer is correct, and we have reversed the legend of Figure 1a and 1b in the revised manuscript.

26. Figure 1c, it should be stated that GUS staining was performed after 48 h of infiltration in the legend.

Reply: Thank you for your suggestion. The reviewer is correct, and we have revised the legend of Figure 1c in the revised manuscript.

27. Figure 2a, describes what each picture represents in each legend.

Reply: Thank you for your suggestion. We have described each picture in the legend of Figure 2a.

28. Figure 2b, add scale bars.

Reply: Thank you for your suggestion. We have added the scale bars in the Figure 2b.

29. Figure 2c, each figure is too small to distinguish. They should be enlarged with a molecular marker.

Reply: Thank you for your suggestion. We have revised each figure of Figure 2a and reuploaded.

30. Figure 5a should include a photo of a WT tomato that grew up at the same time.

Reply: Thank you for your suggestion. We have revised Figure 5a and added the picture of WT tomato.

31. Figure 5b, pointed in Figure 2c, the figure should be enlarged to clearly show the size of the amplified bands and which sample is in each lane.

Reply: Thank you for your suggestion. We have revised each figure of Figure 5 and

reuploaded.

32. Figure 5c, the bar graph in the upper right corner, the description of the vertical axis is missing. "Relative expression of SgEPH2"?

Reply: Thank you for your suggestion. We have revised each figure of Figure 5 and reuploaded. The description of the vertical axis is added.

Editor's Comment: Authors only obtained a single independent transgenic line of the sweet cucumber (Lines 376-377). Results based on only a single independent transgenic line are almost never acceptable.

Reply: On this question, our answer is the following:

1) The paper reported the application of multi-gene vector in cucumber and tomato, but we also emphasized the success of four plant species. Except for transgenic cucumber, other three species obtained multiple transgenic plants. For the lower cucumber transformation efficiency mentioned by the reviewer, to be honest, to solve this problem, we are optimizing this multi-gene transformation system, such as, the screening of more cucumber varieties for transformation, more selected plasmid vectors, better promoters, multi-gene assembly technology, plasmid transformation method, target genes codon optimization, reducing repeated sequence, and so on. And to establish the multiple gene genetic transformation in plants, a lot of systematically optimized work need to be investigated. We think it is a systematic research work to be reported in the future. However, in order to publish innovative work in timer, we choose to publish the positive results obtained so far in "communications biology", and the subsequent research work will be reported in the future. Thank you for your important comments.

2) The paper reported the application of multi-gene vector in cucumber and tomato, but we also emphasized the success of four plant species. Except for transgenic cucumber, other three species obtained multiple transgenic plants. For the lower cucumber transformation efficiency mentioned by the reviewer, to be honest, to solve this problem, we are optimizing this multi-gene transformation system, such as, the screening of more cucumber varieties for transformation, more selected plasmid vectors, better promoters, multi-gene assembly technology, plasmid transformation method, target genes codon optimization, reducing repeated sequence, and so on. And to establish the multiple gene genetic transformation in plants, a lot of systematically optimized work need to be investigated. We think it is a systematic research work to be reported in the future.

However, in order to publish innovative work in timer, we choose to publish the positive results obtained so far in “communications biology”, and the subsequent research work will be reported in the future. Finally, thank you for your important comments.

Reviewers' comments:

Reviewer #3 (Remarks to the Author):

Dear Authors.

Thank you for the authors' responses to my comments. These results seem relatively clear; however, several assertions are still puzzling. In addition, I found that some references were incorrect or inappropriate. The authors should carefully check the references they cited.

[Major comments].

1. L397-398: Were some genes introduced in the other 4 transgenic individuals, or were no other genes introduced at all? The detailed data about it is what I meant that "a complete picture of the transformation experiment." The reason why I think these data is important is 1) it is a challenging experiment to introduce 6 genes in one step-transformation, 2) it is often said that when the T-DNA region is too long, only a part of it is inserted into the genome, especially DNA fragments near the LB side are particularly prone to drop out. Additionally, the authors constructed the vector with multiple 35S promoters placed side by side, which a loop-out recombination event might be happen. The reviewer thinks of determining what happens with the other 4 transgenic individuals is important. The reviewer recommends that once again a table be made to show a complete picture of the transformation experiment.

2. L581: Ref 56 gives a clear mechanism for dwarfish: the constant expression of phytoene synthase has caused a metabolic disturbance in GA. The reviewer thinks that the authors should consider the same point of view in this manuscript. Although the position effect of T-DNA insertion is mentioned in the discussion part, the reviewer does not feel fully support it. Is the position effect of T-DNA insertion happen in 4 individual transgenic lines coincidentally? Is there any other reason for dwarfish in the four independent tomato transgenic lines? Is MIII harmful to tomatoes or not? There is a lack of consideration from a metabolic point of view. Additionally, why did the growth stunt in the tomato transgenic line while the cucumber U1 line grew as a wild type? The reviewer feels this question has not been adequately answered.

[Minor comments]

3. L112: Ref 32 is not about golden rice. Please cite the correct paper.

4. L223 "5/8" should be written as "5 or 8".

5. L305: 0.5-0.6 "mm" Is this a mistake for 0.5-0.6 cm? Please check.

6. L414-415: The authors mentioned the effect of the 2A peptide. This sentence corresponds to the discussion section. Did it also happen in the tomato transformants? Please mention it in the discussion section as well.

7. Figure 2 a: "seedling" should be corrected to "sterilized seeds" or "germinating seeds".

8. Figure 2 c: Gene names are missing. Please specify which gene is amplified.

9. Figure 5, Figure 6, etc: In what part of the tomato was the amount of mogrosides measured? In fruits? Since it is not stated, please state it accordingly.

10. L584, Ref 58: From the text of ref 58, it seems that there is no mention of physiological disorders or plant resistance due to T-DNA insertion. Could you please check? (The abstract also clearly states that no gross phenotypical differences were observed between high-flavonol transgenic and control lines.)

Dear reviewers,

We feel great thanks for your professional review work on our manuscript entitled "A new model of breeding sweet vegetables: synthesizing diverse mogrosides in cucumber and tomato by genetic manipulation". As you are concerned, there are several problems that need to be addressed. According to your nice suggestions, we have made extensive corrections to our previous draft, the detailed corrections are listed below.

Reviewer #3:

Response to comment:

[Major comments].

1. L397-398: Were some genes introduced in the other 4 transgenic individuals, or were no other genes introduced at all? The detailed data about it is what I meant that "a complete picture of the transformation experiment." The reason why I think these data is important is 1) it is a challenging experiment to introduce 6 genes in one step-transformation, 2) it is often said that when the T-DNA region is too long, only a part of it is inserted into the genome, especially DNA fragments near the LB side are particularly prone to drop out. Additionally, the authors constructed the vector with multiple 35S promoters placed side by side, which a loop-out recombination event might be happen. The reviewer thinks of determining what happens with the other 4 transgenic individuals is important. The reviewer recommends that once again a table be made to show a complete picture of the transformation experiment.

Reply: Thank you for your advice. The detailed data about cucumber transformation experiment has been listed in the Supplementary Table S1. And we have stated in the **Generation of mogrosides-accumulating cucumber.**

2. L581: Ref 56 gives a clear mechanism for dwarfish: the constant expression of phytoene synthase has caused a metabolic disturbance in GA. The reviewer thinks that the authors should consider the same point of view in this manuscript. Although the

position effect of T-DNA insertion is mentioned in the discussion part, the reviewer does not feel fully support it. Is the position effect of T-DNA insertion happen in 4 individual transgenic lines coincidentally? Is there any other reason for dwarfish in the four independent tomato transgenic lines? Is MIII harmful to tomatoes or not? There is a lack of consideration from a metabolic point of view. Additionally, why did the growth stunt in the tomato transgenic line while the cucumber U1 line grew as a wild type? The reviewer feels this question has not been adequately answered.

Reply: In our paper, the morphological observation results of transgenic tomato plants revealed that serious dwarfish was observed in the four independent tomato transgenic lines, but no obvious morphological changes were found in the transgenic cucumber plants. This result suggests that mogrosides can affect plant growth, which may produce metabolic toxic effects on transgenic tomato lines. The possible explanation for the absence of dwarfing in cucumber is that both cucumber and *Siraitia grosvenorii* are cucurbitaceae plants, and in cucumber itself there are many cucurbitane triterpene saponin such as cucurbitacin B and cucurbitacin C, and mogrosides are also triterpene saponin, in this case, there is no toxicity effect on the transgenic cucumber lines. Moreover, according to the previous researches, most studies have proven that transgenic engineering has caused a metabolic disturbance in GA, which leads to the dwarfish in the transgenic tomato lines. (Ref 56; Ref 57), Therefore, we speculated that the dwarfish of transgenic tomato in this study may be correlated to the changes in gibberellin metabolism, but the specific reasons need to be verified by subsequent experiments.

[Minor comments]

3. L112: Ref 32 is not about golden rice. Please cite the correct paper.

Reply: Thank you for pointing this out. We have cited the right Ref 32 in the revised manuscript.

4. L223 “5/8” should be written as “5 or 8”.

Reply: Thank you for pointing this out. “5/8” has been corrected on “5 or 8”.

5. L305: 0.5-0.6 “mm” Is this a mistake for 0.5-0.6 cm? Please check.

Reply: We feel sorry for our carelessness. In our resubmitted manuscript, the error is revised. We have revised the 0.5-0.6 “mm” with 0.5-0.6 cm.

6. L414-415: The authors mentioned the effect of the 2A peptide. This sentence corresponds to the discussion section. Did it also happen in the tomato transformants? Please mention it in the discussion section as well.

Reply: Thank you for pointing this out. We have stated in the **Generation of MIII-accumulating tomato**.

7. Figure 2 a: “seedling” should be corrected to “sterilized seeds” or “germinating seeds”.

Reply: Thank you for pointing this out. Sterilized seeds have been used in the Figure 2a.

8. Figure 2 c: Gene names are missing. Please specify which gene is amplified.

Reply: Thank you for pointing this out. We have revised the Figure 2c and reuploaded. The gene names are added.

9. Figure 5, Figure 6, etc: In what part of the tomato was the amount of mogrosides measured? In fruits? Since it is not stated, please state it accordingly.

Reply: Thank you for pointing this out. Fruits of transgenic tomatoes have been used for the quantification of mogrosides content and we have stated it accordingly.

10. L584, Ref 58: From the text of ref 58, it seems that there is no mention of physiological disorders or plant resistance due to T-DNA insertion. Could you please check? (The abstract also clearly states that no gross phenotypical differences were observed between high-flavonol transgenic and control lines.)

Reply: Thank you for your suggestion. The reviewer is correct, and we have replaced the Ref 58 in the revised manuscript.

REVIEWERS' COMMENTS:

Reviewer #3 (Remarks to the Author):

Dear authors,

Significant efforts were made by the authors to address the reviewers' comments. This manuscript has improved since it was last submitted and was also properly corrected.

About the authors' reply to Major comment 2

The authors' interpretations seem to be reasonable. The reviewer think it would be better to comment for the speculation that growth was inhibited in the tomato transgenic line, but not in the cucumber U1 line in their Discussion session. In addition, it would be informative to add the authors' speculation that the reason for the dwarfish of transgenic tomato may be correlated to the changes in gibberellin metabolism in the Discussion session.

Figure 2a(2): I apologize for the confusion in my explanation. I meant to say that "seedling" in Figure 2a(1) should be corrected to "sterilized seeds." Could you correct "4-day-old sterilized seeds" in Figure 2a-(2) to "4-day-old seedlings" ?

Dear reviewers,

We feel great thanks for your professional review work on our manuscript entitled "Heterologous mogrosides biosynthesis in cucumber and tomato by genetic manipulation". As you are concerned, there are several problems that need to be addressed. According to your nice suggestions, we have made extensive corrections to our previous draft, the detailed corrections are listed below.

Reviewer #3:

Response to comment:

1. The authors' interpretations seem to be reasonable. The reviewer think it would be better to comment for the speculation that growth was inhibited in the tomato transgenic line, but not in the cucumber U1 line in their Discussion session. In addition, it would be informative to add the authors' speculation that the reason for the dwarfish of transgenic tomato may be correlated to the changes in gibberellin metabolism in the Discussion session.

Reply: Thank you for your suggestion. We have modified the Discussion part in the revised manuscript.

2. Figure 2a(2): I apologize for the confusion in my explanation. I meant to say that "seedling" in Figure 2a(1) should be corrected to "sterilized seeds." Could you correct "4-day-old sterilized seeds" in Figure 2a-(2) to "4-day-old seedlings" ?

Reply: Thank you for pointing this out. We have been corrected.